# Learning to Mitigate Externalities: the Coase Theorem with Hindsight Rationality

**Antoine Scheid[1]**             **Aymeric Capitaine[1]**

**Etienne Boursier[2]**    **Eric Moulines[1]**    **Michael I. Jordan[3,4]**    **Alain Durmus[1]**

[1] Centre de Mathématiques Appliquées – CNRS – École polytechnique – Palaiseau, 91120, France
[2] INRIA Saclay, Université Paris Saclay, LMO - Orsay, 91400, France
[3] University of California, Berkeley
[4] Inria, Ecole Normale Supérieure, PSL Research University - Paris, 75, France

## Abstract

In economic theory, the concept of externality refers to any indirect effect resulting from an interaction between players that affects the social welfare. Most of the models within which externality has been studied assume that agents have perfect knowledge of their environment and preferences. This is a major hindrance to the practical implementation of many proposed solutions. To address this issue, we consider a two-player bandit setting where the actions of one of the players affect the other player and we extend the Coase theorem [Coase, 2013]. This result shows that the optimal approach for maximizing the social welfare in the presence of externality is to establish property rights, i.e., enable transfers and bargaining between the players. Our work removes the classical assumption that bargainers possess perfect knowledge of the underlying game. We first demonstrate that in the absence of property rights, the social welfare breaks down. We then design a policy for the players which allows them to learn a bargaining strategy which maximizes the total welfare, recovering the Coase theorem under uncertainty.

## 1   Introduction

The concept of *externality* is used in economics to capture phenomena that impact the global welfare stemming from economic interactions without any compensation [Buchanan and Stubblebine, 2006, Shah et al., 2018]. Externality is generally considered as market failure since they result in a loss of collective welfare. Given its practical importance [Dahlman, 1979, Greenfield et al., 2009], mechanisms that characterize and mitigate externalities are central to modern economic thinking.

A first approach to tackle the adverse effects of externalities in modern economics was based on quotas and taxation [see, e.g., Pigou, 2017, and the references therein]. However, Coase's theorem [Coase, 2013] shows that in the presence of well-defined property rights and low transaction costs, parties affected by externalities can privately negotiate efficient solutions, and recover a welfare efficient allocation through transfers and bargaining.

Throughout the paper, we use the following simple example to illustrate our results.

**Example 1.** *Consider two firms 1 (upstream) and 2 (downstream) respectively producing quantities $q_1 \geqslant 0$ and $q_2 \geqslant 0$ of a good sold at a fixed price $p > 0$. They incur strictly increasing and convex costs, captured by the differentiable cost functions $c_1 : q_1 \mapsto c_1(q_1)$ and $c_2 : q_2 \mapsto c_2(q_2)$, satisfying*

$c_1(0) = 0$ and $c_2(0) = 0$. We assume that firm 1 exerts on firm 2 a constant externality $\alpha > 0$ per unit produced. In other words, their profit functions (or utilities) are given by

$$\pi_1(q_1) = pq_1 - c_1(q_1) \quad and \quad \pi_2(q_1, q_2) = pq_2 - c_2(q_2) - \alpha q_1 \ .$$

One simple concrete illustration consists in an upstream firm emitting pollutants that reduce the downstream firm's production. If the upstream firm owns the property rights, it may receive a payment from the downstream one to reduce its output and thereby pollution. On the other hand, if the downstream firm owns the property right, it may require compensation from the upstream firm to allow its operation.

The Coase theorem demonstrates that in both cases with appropriate property rights, the resulting levels of production would be welfare efficient. The theorem is typically explained in textbooks under the assumption that the players have perfect knowledge of their own utility or profit function, as well as that of others. However, this assumption is unlikely to hold in real-world scenarios, where players have to learn about their own preferences and those of their competitors.

This example is, of course, a simplification of real world scenarios. For example, Abildtrup et al. [2012] consider a more complex setting to model the interaction between farmers and waterworks in Denmark where the farmers have the property rights whereas the waterworks can pay to reduce pollution. In particular, they demonstrate the failure of the theorem, attributing it to the breakdown of the main assumptions: no transaction costs, maximizing behaviors and perfect information, with an important focus on strategic behaviors and the asymmetry of information. We restore the latter in our work and provide foundations for the theorem to hold in more realistic scenarios.

*A key question is whether the Coase theorem holds when players learn their preferences over time.*

We investigate this question within the framework of a multi-armed bandit learning. We build upon recent works that extend the classical bandit setting to economics in which there are two players interacting via principal-agent protocols [Dogan et al., 2023b,a, Scheid et al., 2024]. This allows us to capture, for example, a version of the two-firm problem where firms are uncertain regarding both their profit functions and the degree of externality on other firms. We represent production decisions in this problem as arms which can be played by the firms at any round over time, with the goal of finding decisions that maximize their rewards. More precisely, we assume that the reward of the upstream firm only depends on its own action, while the reward of the downstream firm depends on both its action and the upstream firm's action. This dependency on both actions allows us to capture externalities.

The property rights in Example 1, or more generally over a bandit instance, amount to giving a firm the possibility of engaging in monetary transfers that influence the arms that are played over time. The owner of the bandit instance will face a problem of bandit learning with transfers with an upstream player who is also learning his preferences.

To account for the efficiency of a policy in this setup, we extend the classical static notion of welfare efficiency to the online setting. We say that a policy is *Welfare efficient* if the social welfare regret is sub-linear. Proving the Coase theorem within our setup therefore boils down to show that if the bandit owner (who is without loss of generality the upstream player in this study[1]) runs a no-regret bandit algorithm to learn and exploit his preferences, the downstream player can then choose an optimal transfer scheme leading to a sub-linear total regret.

Our contributions are as follows:

- We show that when an upstream agent exerts externality on a downstream agent, in the absence of property right, the social welfare breaks down. Put differently, no joint policy of the agents can be welfare efficient.
- We then introduce property rights and show how it affects the game. In this case, bargaining and transfers are available to the players. We propose a policy for the downstream player that leads to welfare efficiency when the upstream player follows any black-box no-regret policy under mild assumption. This solution addresses the breakdown issue at equilibrium. Put together, we show an online version of the *Coase theorem*.

---

[1]The fact that efficiency is restored whoever is given the property rights is known as the *invariance* property of the Coase theorem.

## 2 Setup and Inefficiency of Externality

### 2.1 Bandit game

We consider a sequential bandit game in which two players (downstream and upstream) simultaneously play actions in a bandit instance for a horizon $T \in \mathbb{N}^\star$. The action set for both players is $\mathcal{A} = \{1, \ldots, K\}, K \in \mathbb{N}^\star$.

The reward distributions of the agents differ. Given a family of distributions $\{\gamma_a : a \in \mathcal{A}\}$ indexed by $\mathcal{A}$, the upstream player's rewards are provided by an i.i.d. family of random variables

$$\{(Z_a(t))_{t \in [T]} : a \in \mathcal{A}\} \ , \quad \text{where } Z_a(t) \sim \gamma_a \quad \text{for any } t \in [T] \text{ and } a \in \mathcal{A} \ .$$

To model the externality exerted by the upstream on the downstream player, we assume that the latter has a reward that depends both on her action and on that of the upstream player. Formally, this is modeled through a family of distributions $\{\nu_{a,b} : a, b \in \mathcal{A}\}$ double-indexed by $\mathcal{A}$ and an i.i.d. family of random variables

$$\{(X_{a,b}(t))_{t \in [T]} : a, b \in \mathcal{A}\} \ , \quad \text{where } X_{a,b}(t) \sim \nu_{a,b}$$

is the reward received by the downstream player at time $t$ if she pulls the arm $b$ and the upstream player pulls the arm $a$.

**Players.** We assume that players are risk-neutral expected-utility maximizers, and we define their expected utilities for any $(a, b) \in \mathcal{A} \times \mathcal{A}$ as

$$v^{\mathrm{up}}(a) = \int z \, \gamma_a(\mathrm{d}z) \in \mathbb{R} \quad \text{and} \quad v^{\mathrm{down}}(a, b) = \int x \, \nu_{a,b}(\mathrm{d}x) \in \mathbb{R} \ .$$

The distributions $(\gamma_a)_{a \in \mathcal{A}}$ and $(\nu_{a,b})_{(a,b) \in \mathcal{A}^2}$ are unknown to both the downstream and the upstream players and they aim to learn the distributions with best mean rewards by sequentially observing samples from $\{(Z_a(t))_{t \in [T]} : a \in \mathcal{A}\}$ and $\{(X_{a,b}(t))_{t \in [T]} : a, b \in \mathcal{A}\}$.

Moreover, we suppose that players are rational in hindsight; that is, they minimize their regret. Formally, the upstream player aims to minimize his regret defined as

$$\mathfrak{R}_{\mathrm{n}}^{\mathrm{up}}(T, \Pi_{\mathrm{n}}^{\mathrm{up}}) = T\mu^{\star,\mathrm{up}} - \mathbb{E}\left[\sum_{t=1}^{T} v^{\mathrm{up}}(A_t)\right] \ , \text{ where } \mu^{\star,\mathrm{up}} = \max_{a \in \mathcal{A}} v^{\mathrm{up}}(a) \ , \tag{1}$$

while the downstream player seeks to minimize her external regret defined as

$$\mathfrak{R}_{\mathrm{n}}^{\mathrm{down}}(T, \Pi_{\mathrm{n}}^{\mathrm{up}}, \Pi_{\mathrm{n}}^{\mathrm{down}}) = \mathbb{E}\left[\sum_{t=1}^{T} \max_{b \in \mathcal{A}} v^{\mathrm{down}}(A_t, b) - v^{\mathrm{down}}(A_t, B_t)\right] \ , \tag{2}$$

where the players' actions $(A_t)_{t \in [T]}, (B_t)_{t \in [T]}$ as well as their policies $\Pi_{\mathrm{n}}^{\mathrm{up}}, \Pi_{\mathrm{n}}^{\mathrm{down}}$ are defined below. Note that the utility of the downstream player also depends on the actions taken by the upstream player, which represents the externality exerted by the upstream player on the downstream player, hence the strategic dimension of our setting. We first consider a game where no property right is defined, so each player is free to pick his preferred arm irrespectively of the other player's choice. This will result in a breakdown of the total utility.

**Policies without property rights.** Consider first the upstream player. Based on a policy $\Pi_{\mathrm{n}}^{\mathrm{up}}$ (for example a no-regret bandit algorithm such as the `Upper Confidence Bounds` algorithm (`UCB`) [Auer, 2002] or the $\varepsilon$-`greedy` algorithm [Robbins, 1952, Langford and Zhang, 2007]), we define his history $(\mathcal{H}_t^{\mathrm{up,n}})_{t \in [T]}$ by induction. We set $\mathcal{H}_0^{\mathrm{up,n}} = \varnothing$ and supposing that $\mathcal{H}_t^{\mathrm{up,n}}$ is defined for $t \in [T]$, then

$$\mathcal{H}_{t+1}^{\mathrm{up,n}} = \mathcal{H}_t^{\mathrm{up,n}} \cup \{A_{t+1}, V_{t+1}, Z_{A_{t+1}}(t+1)\} \ ,$$

where $(V_s)_{s \in \mathbb{N}^\star}$ is a family of independent uniform random variables in $[0, 1]$, allowing for randomization in the policy, and $A_{t+1}$ is provided by $\Pi_{\mathrm{n}}^{\mathrm{up}}$, following $\Pi_{\mathrm{n}}^{\mathrm{up}} : (V_{t+1}, \mathcal{H}_t^{\mathrm{up,n}}) \mapsto A_{t+1}$.

Second, consider the downstream player and an algorithm $\Pi_{\mathrm{n}}^{\mathrm{down}}$ (specifically a no-regret bandit algorithm). We define her history $(\mathcal{H}_t^{\mathrm{down,n}})_{t \in [T]}$ by induction. We set $\mathcal{H}_0^{\mathrm{down,n}} = \varnothing$ and supposing that $\mathcal{H}_t^{\mathrm{down,n}}$ is defined for $t \in [T]$, then

$$\mathcal{H}_{t+1}^{\mathrm{down,n}} = \mathcal{H}_t^{\mathrm{down,n}} \cup \{A_{t+1}, B_{t+1}, U_{t+1} X_{A_{t+1}, B_{t+1}}(t+1)\} \ ,$$

where $(U_s)_{s \in \mathbb{N}^\star}$ is a family of independent uniform random variables in $[0, 1]$ allowing for randomization in the policy and $B_{t+1}$ is provided by $\Pi_n^{\text{down}}$, following $\Pi_n^{\text{down}} : (U_{t+1}, \mathcal{H}_t^{\text{down,n}}) \mapsto B_{t+1}$.

**Welfare efficiency.** We now introduce the notion of *Welfare efficiency* for our setup. The global utility, or *social welfare*, of the players at round $t$ is defined as $v^{\text{up}}(A_t) + v^{\text{down}}(A_t, B_t)$. We define the *socially optimal action* $(a^{\text{sw}}, b^{\text{sw}}) \in \mathcal{A} \times \mathcal{A}$ of the game as

$$(a^{\text{sw}}, b^{\text{sw}}) \in \text{argmax}_{a,b \in \mathcal{A}} \ v^{\text{up}}(a) + v^{\text{down}}(a, b) , \tag{3}$$

as well as the *global regret* (or *social welfare regret*) associated with policies $\Pi_n^{\text{up}}$ and $\Pi_n^{\text{down}}$ as

$$\mathfrak{R}^{\text{sw}}(T, \Pi_n^{\text{up}}, \Pi_n^{\text{down}}) = T \left( v^{\text{up}}(a^{\text{sw}}) + v^{\text{down}}(a^{\text{sw}}, b^{\text{sw}}) \right) - \sum_{t=1}^{T} \mathbb{E}\left[ v^{\text{up}}(A_t) + v^{\text{down}}(A_t, B_t) \right] . \tag{4}$$

Then, the joint policies $\Pi_n^{\text{up}}$ and $\Pi_n^{\text{down}}$ for the players are said to be *Welfare efficient* if

$$\lim_{T \to +\infty} \mathfrak{R}^{\text{sw}}(T, \Pi_n^{\text{up}}, \Pi_n^{\text{down}})/T = 0 .$$

Intuitively, this condition implies that the frequency of the socially optimal action $(a^{\text{sw}}, b^{\text{sw}})$ tends to 1 as $T$ goes to infinity. In this sense, it mimics the usual, static Welfare efficiency criterion. As we will see, $(\Pi_n^{\text{up}}, \Pi_n^{\text{down}})$ is typically not *Welfare efficient* when there is a disalignment in the game between the players' individual interests based on their rationality and the social welfare.

## 2.2 Inefficiency without property rights

We first present a result that captures the adverse consequence of externality on social welfare. The upstream player does not take into account the indirect cost incurred by the downstream player when he chooses his action. This drives the social welfare away from its optimal level. We illustrate this fact within our simple bilateral externality example.

**Example 1** (continuing from p. 1)**.** *We show that the competitive outcome, where each firm maximizes its profit independently, is not welfare efficient in the presence of externality. Define the social welfare as the function*

$$W : (q_1, q_2) \mapsto \pi_1(q_1) + \pi_2(q_1, q_2) = p(q_1 + q_2) - (c_1(q_1) + c_2(q_2)) - \alpha q_1 . \tag{5}$$

*By definition, the welfare efficient outcome $(q_1^\star, q_2^\star) \in \mathbb{R}_+^2$ satisfies $W(q_1^\star, q_2^\star) \geqslant W(q_1, q_2)$ for any $(q_1, q_2) \in \mathbb{R}_+^2$. Since $W$ is differentiable and strictly concave, $(q_1^\star, q_2^\star)$ is uniquely defined by the condition $\nabla W(q_1^\star, q_2^\star) = 0$, that is*

$$c_1'(q_1) - \alpha = p \quad and \quad c_2'(q_2) = p . \tag{6}$$

*Note that at the welfare efficient optimum, firm 1 does not equalize marginal cost with marginal profit, but produces less to account for the negative effect of externality on firm 2. We now characterize the competitive outcome $(q_1', q_2') \in \mathbb{R}_+^2$. Since $\pi_1$ and $\pi_2$ are differentiable and strictly concave, $(q_1', q_2')$ satisfies*

$$c_1'(q_1') = p \quad and \quad c_2'(q_2') = p . \tag{7}$$

*For the competitive outcome to be welfare efficient, we require, by Equation (6) and Equation (7),*

$$c_1'(q_1') - \alpha = c_1'(q_1'), \quad that \ is \quad \alpha = 0 .$$

*This proves that whenever there are externalities, no competitive outcome is efficient.*

We now show that in our model, when there is no property right and under mild assumptions, no achievable policy is welfare efficient whenever there is a misalignment between the players' interests and the social welfare. The upstream player's policy $\Pi_n^{\text{up}}$ is said to be *no-regret* if $\lim_{T \to +\infty} \mathfrak{R}_n^{\text{up}}(T, \Pi_n^{\text{up}})/T = 0$, where $\mathfrak{R}_n^{\text{up}}$ is defined in (1).

**Theorem 2.** *Suppose that $\text{argmax}_{a \in \mathcal{A}} v^{\text{up}}(a)$ is the singleton $\{a_\star^{\text{u}}\}$ and that*

$$v^{\text{up}}(a^{\text{sw}}) + v^{\text{down}}(a^{\text{sw}}, b^{\text{sw}}) - v^{\text{up}}(a_\star^{\text{u}}) + v^{\text{down}}(a_\star^{\text{u}}, b) > 0 , \tag{8}$$

*for any $b \in \mathcal{A}$. In the absence of property rights and when the upstream player runs any no-regret policy $\Pi_n^{\text{up}}$, we have $\mathfrak{R}^{\text{sw}}(T, \Pi_n^{\text{up}}, \Pi_n^{\text{down}}) = \Omega(T)$. Therefore, $\mathfrak{R}^{\text{sw}}(T, \Pi_n^{\text{up}}, \Pi_n^{\text{down}}) = \Omega(T)$ and $(\Pi_n^{\text{up}}, \Pi_n^{\text{down}})$ is not welfare efficient.*

Condition (8) in Theorem 2 represents the unalignment between the upstream player's preference and the optimal choice from a social welfare point of view. Note that the upstream and downstream players can both have an $o(T)$ external regret, while the social welfare regret still grows linearly with $T$ because of the unfavorable interactions between their policies.

## 3 Online Property Game with Bargaining Players

### 3.1 Online Property Game

We now consider the same repeated game in the form of a *property game* where one of the players possesses the bandit instance (*upstream player*). As in the original setup of Coase [2013], the other player (*downstream player*) will provide the bandit owner with transfers to incentivize him to choose some specific action and influence the outcome of the game in her favor.

We show in Appendix B that our method applies similarly when property rights are given to the upstream player rather than the downstream player. Hence, there is no loss of generality in considering the aforementioned framework. In this sense, we recover the *invariance property* of the Coasean bargaining [Mas-Colell et al., 1995].

**Example 1** (continuing from p. 1). *We now illustrate how Coasean bargaining re-instaures efficiency. Suppose without loss of generality that property rights are such that firm 2 can pay $\tau \in \mathbb{R}_+$ to firm 1 for it to operate at a level $\tilde{q}_1$. Profits become*

$$\bar{\pi}_2 : (q_1, q_2, \tau, \tilde{q}_1) \mapsto \pi_2(q_1, q_2) - \mathbb{1}_{\{q_1 = \tilde{q}_1\}} \tau \quad and \quad \bar{\pi}_1 : (q_1, \tau, \tilde{q}_1) \mapsto \pi(q_1) + \mathbb{1}_{\{q_1 = \tilde{q}_1\}} \tau.$$

*Consider the competitive outcome $(q_1, q_2, \tau, \tilde{q}_1) \in \mathbb{R}_+^4$ which satisfies*

$$q_1 = q_1(\tau, \tilde{q}_1) \in \arg\max_{q_1' \geqslant 0} \bar{\pi}_1(q_1', \tau, \tilde{q}_1) \quad and$$

$$\bar{\pi}_2(q_1(\tau, \tilde{q}_1), q_2, \tau, \tilde{q}_1) = \max_{q_2', \tau', \tilde{q}_1'} \bar{\pi}_2(q_1(\tau', \tilde{q}_1'), q_2', \tau', \tilde{q}_1').$$

*The condition on $q_1$ accounts for the rationality of the firm 1 and the fact that its choice depends on the payment $(\tau, \tilde{q}_1)$. Obviously, the optimal solution is reached for $\tilde{q}_1 = q_1$ and $\tau = \max_{q'} \pi_1(q') - \pi_1(\tilde{q}_1)$. Plugging this back in the expression of $\bar{\pi}_2$ then yields*

$$(q_1, q_2) = \mathrm{argmax}_{q_1', q_2'} \pi_1(q_1') + \pi_2(q_2') = \mathrm{argmax}_{q_1', q_2'} W(q_1', q_2') \, ,$$

*so the competitive outcome $(q_1, q_2)$ is welfare efficient.*

The transfers at each step can be interpreted as a contract between two players [see, e.g., Bolton and Dewatripont, 2004, Salanié, 2005, for general contract theory] and providing the right amount of incentives relates to adjusting a contract in an online setting [see Dütting et al., 2019, Guruganesh et al., 2021, Zhu et al., 2022, Fallah and Jordan, 2023, Guruganesh et al., 2024, Ananthakrishnan et al., 2024, for learning-based perspectives about contracts].

Similarly to Example 1, we modify the players' policies to now account for the transfer $\tau(t)$ that the downstream player offers at round $t$ to the upstream player if he picks action $\tilde{a}_t$. The downstream player's policy at round $t$ does not only output an arm $B_t$ but now a triple $(\tilde{a}_t, \tau(t), B_t)$, where $B_t$ is the arm that she should play and $\tilde{a}_t$ is the arm on which a transfer $\tau(t)$ is offered to the upstream player. On the upstream player's side, the policy still outputs an arm $A_t$ to play but also takes as an input the incentive $(\tilde{a}_t, \tau(t))$. In addition, the instantaneous utility of the upstream player becomes $Z_{A_t}(t) + \mathbb{1}_{\tilde{a}_t}(A_t)\tau(t)$, whereas the downstream player receives $X_{A_t, B_t}(t) - \mathbb{1}_{\tilde{a}_t}(A_t)\tau(t)$.

**Policies with property rights.** Based on policies $\Pi_p^{\mathrm{up}}$ for the upstream player and $\Pi_p^{\mathrm{down}}$ for the downstream player, we define their histories $(\mathcal{H}_t^{\mathrm{up,p}})_{t \in [T]}$ and $(\mathcal{H}_t^{\mathrm{down,p}})_{t \in [T]}$ by induction. We set $\mathcal{H}_0^{\mathrm{up,p}} = \varnothing$, $\mathcal{H}_0^{\mathrm{down,p}} = \varnothing$ and supposing that $\mathcal{H}_t^{\mathrm{up,p}}, \mathcal{H}_t^{\mathrm{down,p}}$ are defined for $t \in [T]$, then

$$\mathcal{H}_{t+1}^{\mathrm{up,p}} = \mathcal{H}_t^{\mathrm{up,p}} \cup \left\{ \tilde{a}_{t+1}, \tau(t+1), A_{t+1}, V_{t+1}, Z_{A_{t+1}}(t+1) \right\}$$

and

$$\mathcal{H}_{t+1}^{\mathrm{down,p}} = \mathcal{H}_t^{\mathrm{down,p}} \cup \left\{ \tilde{a}_{t+1}, \tau(t+1), A_{t+1}, B_{t+1}, U_{t+1}, X_{A_{t+1}, B_{t+1}}(t+1) \right\} \, ,$$

where $(V_s)_{s\in\mathbb{N}^\star}$, $(U_s)_{s\in\mathbb{N}^\star}$ are two families of independent uniform random variables in $[0,1]$ allowing for randomization in the policies, and the remaining quantities are given by $\Pi_{\mathrm{p}}^{\mathrm{down}} : (U_{t+1}, \mathcal{H}_t^{\mathrm{down,p}}) \mapsto (\tilde{a}_{t+1}, \tau(t+1), B_{t+1})$ and $\Pi_{\mathrm{p}}^{\mathrm{up}} : (\tilde{a}_{t+1}, \tau(t+1), V_{t+1}, \mathcal{H}_t^{\mathrm{up,p}}) \mapsto A_{t+1}$.

**Players' goal.** Given a transfer $\tau$ from the downstream to the upstream player on arm $\tilde{a}$, actions $a$ and $b$ respectively are chosen by the upstream and the downstream player, the upstream player's expected utility reads $v^{\mathrm{up}}(a) + \mathbb{1}_{\tilde{a}}(a)\tau$ while the downstream player's expected utility is $v^{\mathrm{down}}(a,b) - \mathbb{1}_{\tilde{a}}(a)\tau$. This defines the upstream player's expected regret for a horizon $T$ as

$$\mathfrak{R}_{\mathrm{p}}^{\mathrm{up}}(T, \Pi_{\mathrm{p}}^{\mathrm{up}}, \Pi_{\mathrm{p}}^{\mathrm{down}}) = \mathbb{E}\left[\sum_{t=1}^{T} \max_{a\in\mathcal{A}}\{v^{\mathrm{up}}(a) + \mathbb{1}_{\tilde{a}_t}(a)\tau(t)\} - (v^{\mathrm{up}}(A_t) + \mathbb{1}_{\tilde{a}_t}(A_t)\tau(t))\right]. \quad (9)$$

Based on the upstream player's utility, the downstream player aims on a single round at proposing an optimal transfer $\tau^{\mathrm{opt}}$ on an arm $a^{\mathrm{opt}} \in \mathcal{A}$ as well as picking an arm $b^{\mathrm{opt}} \in \mathcal{A}$ which solves

$$\begin{aligned} &\text{maximize } (a,b,\tau) \mapsto v^{\mathrm{down}}(a,b) - \tau \\ &\text{such that } \tau \in \mathbb{R}_+, b \in \mathcal{A}, a \in \mathrm{argmax}_{a'\in\mathcal{A}}\{v^{\mathrm{up}}(a') + \mathbb{1}_a(a')\tau\}\,. \end{aligned} \quad (10)$$

Her regret for any horizon $T$ is defined as

$$\mathfrak{R}_{\mathrm{p}}^{\mathrm{down}}(T, \Pi_{\mathrm{p}}^{\mathrm{up}}, \Pi_{\mathrm{p}}^{\mathrm{down}}) = T\mu^{\star,\mathrm{down}} - \mathbb{E}\left[\sum_{t=1}^{T} v^{\mathrm{down}}(A_t, B_t) - \mathbb{1}_{\tilde{a}_t}(A_t)\tau(t)\right], \quad (11)$$

where we define $\mu^{\star,\mathrm{down}} = v^{\mathrm{down}}(a^{\mathrm{opt}}, b^{\mathrm{opt}}) - \tau^{\mathrm{opt}}$ as the optimal utility she can aim for. We can see that the downstream player's influence is exerted through her action choice $B_t$ as well as through transfers which enable her to influence the upstream player's actions. Hence, the notion of external regret is obsolete here. The game has now the form of a repeated *Stackelberg game* [Von Stackelberg, 2010].

**Lemma 1.** *Recall that $\mu^{\star,\mathrm{down}}$ is the downstream player's optimal reward as defined as a solution of* (10). *We have $\mu^{\star,\mathrm{down}} = \max_{a,b\in\mathcal{A}}\{v^{\mathrm{down}}(a,b) + v^{\mathrm{up}}(a)\} - \max_{a'\in\mathcal{A}}\{v^{\mathrm{up}}(a')\}$, as well as $(a^{\mathrm{opt}}, b^{\mathrm{opt}}) = (a^{\mathrm{sw}}, b^{\mathrm{sw}})$ and $\mu^{\star,\mathrm{up}} + \mu^{\star,\mathrm{down}} = v^{\mathrm{up}}(a^{\mathrm{sw}}) + v^{\mathrm{down}}(a^{\mathrm{sw}}, b^{\mathrm{sw}}) = \max_{a,b\in\mathcal{A}}\{v^{\mathrm{up}}(a) + v^{\mathrm{down}}(a,b)\}$, where $\mu^{\star,\mathrm{up}}$ is defined in Equation* (1). *Moreover, for any integer $T \in \mathbb{N}^\star$, and policies $\Pi_{\mathrm{p}}^{\mathrm{up}}$, $\Pi_{\mathrm{p}}^{\mathrm{down}}$, we have that*

$$\mathfrak{R}^{\mathrm{sw}}(T, \Pi_{\mathrm{p}}^{\mathrm{up}}, \Pi_{\mathrm{p}}^{\mathrm{down}}) \leqslant \mathfrak{R}_{\mathrm{p}}^{\mathrm{up}}(T, \Pi_{\mathrm{p}}^{\mathrm{up}}, \Pi_{\mathrm{p}}^{\mathrm{down}}) + \mathfrak{R}_{\mathrm{p}}^{\mathrm{down}}(T, \Pi_{\mathrm{p}}^{\mathrm{up}}, \Pi_{\mathrm{p}}^{\mathrm{down}})\,.$$

This lemma has an interesting economic interpretation: if both players individually seek for their own interest within this online property game, they will together converge towards the optimal global utility. Individual rationality moves the outcome of the game towards the optimal social welfare. The transfers allow the players to align their goals and share the global reward, in line with the Coase theorem. Consequently, if both players run no-regret policies $\Pi_{\mathrm{p}}^{\mathrm{up}}$ and $\Pi_{\mathrm{p}}^{\mathrm{down}}$, the social welfare regret will also be in $o(T)$. The rest of the paper shows that such no-regret policies exist. To this end, we introduce the following assumptions.

Without loss of generality, we assume that the upstream player's utility is rescaled and shifted, which corresponds to the following assumption on the reward distribution $(\gamma_a)_{a\in\mathcal{A}}$ in $\mathbb{R}$.

**H1.** *For any $a \in \mathcal{A}$, we have $v^{\mathrm{up}}(a) \in [0,1]$.*

We now make a high probability bound assumption on the upstream player's regret.[2]

**H2.** *There exist $\mathrm{C}, \zeta > 0, \kappa \in [0,1)$ such that for any $s,t \in [T]$ with $s+t \leqslant T$, any $\{\tau_a\}_{a\in[K]} \in \mathbb{R}_+^K$ and any policy $\Pi_{\mathrm{p}}^{\mathrm{down}}$ that offers almost surely a transfer $(\tilde{a}_l, \tau(l)) = (\tilde{a}_l, \tau_{\tilde{a}_l})$ for any $l \in \{s+1, \ldots, s+t\}$, the batched regret of the upstream player following $\Pi_{\mathrm{p}}^{\mathrm{up}}$ satisfies, with probability at least $1 - t^{-\zeta}$,*

$$\sum_{l=s+1}^{s+t} \max_{a\in\mathcal{A}}\{v^{\mathrm{up}}(a) + \mathbb{1}_{\tilde{a}_l}(a)\tau_{\tilde{a}_l}\} - (v^{\mathrm{up}}(A_l) + \mathbb{1}_{\tilde{a}_l}(A_l)\tau_{\tilde{a}_l}) \leqslant \mathrm{C}t^{\kappa}\,.$$

---

[2]A similar assumption is made in the work of Donahue et al. [2024] but with a stronger instantaneous regret bound which does not encompass the UCB's regret bound.

The constraint on the downstream player's algorithm $\Pi_{\mathrm{p}}^{\mathrm{down}}$ enforces constant incentives associated with any arm $a \in \mathcal{A}$ within the batch, while the incentivized actions $(\tilde{a}_l)_{l \in \{s+1, \ldots, s+t\}}$ may change. Proposition 2 in Appendix C shows that an adaptation of UCB taking account the incentives satisfies **H2** with $\mathrm{C} = 8\sqrt{K \log(KT^3)}$, $\kappa = 1/2$ and $\zeta = 2$. Note that usual bandit algorithms such as AAE, ETC or EXP-IX also satisfy the assumption [see, e.g., Donahue et al., 2024, Lattimore and Szepesvári, 2020].

## 3.2 Downstream player's procedure

We fix the policy $\Pi_{\mathrm{p}}^{\mathrm{up}}$ which can be any algorithm satisfying **H2** for the upstream player and introduce the algorithm BELGIC (Bandits and Externalities for a Learning Game with Incentivized Coase) which provides a policy achieving sub-linear regret for the downstream player. It can be seen as an online bargaining strategy to mitigate externalities. Simply put, BELGIC unfolds in two steps. First note that for any action $a \in \mathcal{A}$, the optimal (lowest) transfer to offer to the upstream player to make him choose $a$ is

$$\tau_a^\star = \max_{a' \in \mathcal{A}} v^{\mathrm{up}}(a') - v^{\mathrm{up}}(a) , \tag{12}$$

as detailed in Appendix A. Therefore, a batched binary search procedure (Algorithm 2) first allows the downstream player to estimate the optimal transfers $\tau_1^\star, \ldots, \tau_K^\star$ with a good precision level of $1/T^\beta$, where $\beta > 0$. More precisely, the downstream player offers a constant incentive $(\tilde{a}, \tau_{\tilde{a}})$ for a batch of time steps of length $\tilde{T} = \lceil T^\alpha \rceil$. The observation of $T_{\tilde{a}}^{\neq}$, the number of steps from the batch for which the upstream player does not pick $\tilde{a}$ allows her to estimate whether $\tau_{\tilde{a}}$ is above or below $\tau_{\tilde{a}}^\star$ and adjust it, following Lemma 2 in Appendix A under the condition that $\alpha, \beta$ satisfy

$$\beta/\alpha < (1 - \kappa) . \tag{13}$$

The procedure needs to be run for $K \lceil T^\alpha \rceil \lceil \log_2 T^\beta \rceil$ rounds since we have to make $\lceil \log_2 T^\beta \rceil$ batches of binary search of length $\lceil T^\alpha \rceil$ on each of the $K$ arms [see Scheid et al., 2024]. This corresponds to the first phase of BELGIC as described in Algorithm 2. At the end of this stage, the estimated transfers $(\hat{\tau}_a)_{a \in \mathcal{A}}$ satisfy the bound in Proposition 1. These are then used to feed the subroutine Bandit-Alg.

**Proposition 1.** *Under **H1** and **H2**, after the first phase of* BELGIC *which consists in* $K \lceil T^\alpha \rceil \lceil \log T^\beta \rceil$ *steps of binary search grouped in* $\lceil \log_2 T^\beta \rceil$ *batches per arm* $a \in \mathcal{A}$, *we have that*

$$\mathbb{P}\Big( \text{for any } a \in \mathcal{A}, \hat{\tau}_a - 4/T^\beta - \mathrm{C}T^{(\kappa-1)/2} \leqslant \tau_a^\star \leqslant \hat{\tau}_a \Big) \geqslant 1 - K \lceil \log_2 T^\beta \rceil / T^{\alpha\zeta} .$$

The additional term $1/T^\beta + \mathrm{C}T^{\kappa-1}$ in $\hat{\tau}_a$ ensures that if **H2** holds, the upstream player necessarily plays the incentivized action $\tilde{a}_t$ at round $t$ with high probability. Lemmas 2 and 6 from Appendix C show how the binary search batches in BELGIC allow us to estimate $\tau_a^\star$ depending on $\tilde{T} - T_a^{\neq}$, the number of times that arm $a$ has been pulled by the upstream player during the batch.

Then, any bandit subroutine Bandit-Alg, such as UCB or $\varepsilon$-greedy, for instance, can be run in a black-box fashion on the shifted bandit instance, where the rewards are shifted by the upper estimated transfers $(\hat{\tau}_a)_{a \in \mathcal{A}}$. The downstream player computes a shifted history $\tilde{\mathcal{H}}_t^{\mathrm{down,p}}$ such that for any $t \leqslant K \lceil T^\alpha \rceil \lceil \log_2 T^\beta \rceil, \tilde{\mathcal{H}}_t^{\mathrm{down,p}} = \varnothing$ and for any $t > K \lceil T^\alpha \rceil \lceil \log_2 T^\beta \rceil$

$$\tilde{\mathcal{H}}_t^{\mathrm{down,p}} = \begin{cases} \{\tilde{a}_t, B_t, \tau(t), U_t, X_{\tilde{a}_t, B_t}(t) - \hat{\tau}_{\tilde{a}_t}\} \cup \tilde{\mathcal{H}}_{t-1}^{\mathrm{down,p}} \text{ if } \tilde{a}_t = A_t \\ \tilde{\mathcal{H}}_{t-1}^{\mathrm{down,p}} \quad \text{otherwise} , \end{cases} \tag{14}$$

which serves to feed Bandit-Alg, following

$$\texttt{Bandit-Alg}: (U_t, \tilde{\mathcal{H}}_{t-1}^{\mathrm{down,p}}) \mapsto (\tilde{a}_t, B_t) \in \mathcal{A} \times \mathcal{A} . \tag{15}$$

For any family of constant incentives $\{\tau_a\}_{a \in \mathcal{A}} \in \mathbb{R}_+^K$, we define $\mathfrak{R}_{\texttt{Bandit-Alg}}(T, \nu, \{\tau_a\}_{a \in \mathcal{A}})$ as the regret for the downstream player's subroutine Bandit-Alg on the bandit instance with shifted means over $T$ rounds, following

$$\mathfrak{R}_{\texttt{Bandit-Alg}}(T, \nu, \{\tau_a\}_{a \in \mathcal{A}}) = T \max_{a,b \in \mathcal{A}^2} \mathbb{E}\big[v_{a,b}^{\mathrm{down}}(1) - \tau_a\big] - \mathbb{E}\left[\sum_{t=1}^{T} v^{\mathrm{down}}(\tilde{a}_t, B_t) - \tau_{\tilde{a}_t}\right] .$$

Note that here, Bandit-Alg aims to maximize the shifted reward $(v^{\mathrm{down}}(a, b) - \tau_a)_{(a,b) \in \mathcal{A}^2}$.

---

**Algorithm 1** BELGIC

---

1: **Input:** Set of actions $\mathcal{A} = [K]$, time horizon $T$, subroutine $\Pi_{\mathrm{p}}^{\mathrm{up}}$, upstream player's regret constants C, $\kappa$, parameters $\alpha$ and $\beta$.
2: Compute $\tilde{\mathcal{H}}_s^{\mathrm{down,p}} = \varnothing$ for any $s \leqslant K\lceil \log_2 T^\beta \rceil \lceil T^\alpha \rceil$.
3: **for** $a \in \mathcal{A}$ **do**
4:      # See Algorithm 2
5:      $\underline{\tau}_a, \overline{\tau}_a =$ Binary Search$(a, \lceil \log_2 T^\beta \rceil, \lceil T^\alpha \rceil, 0, 1)$
6: **end for**
7: For any action $a \in \mathcal{A}$, $\hat{\tau}_a = \overline{\tau}_a + 1/T^\beta + CT^{(\kappa-1)/2}$.
8: **for** $t = K\lceil T^\alpha \rceil \lceil \log_2 T^\beta \rceil + 1, \ldots, T$ **do**
9:      Get recommended actions by Bandit-Alg on the $\mathcal{A} \times \mathcal{A}$ bandit instance, $(\tilde{a}_t, B_t) =$ Bandit-Alg$(U_t, \tilde{\mathcal{H}}_{t-1}^{\mathrm{down,p}})$.
10:      Offer a transfer $\hat{\tau}_{\tilde{a}_t}$ on action $\tilde{a}_t$, nothing for any other action $a' \in \mathcal{A}$ and play action $B_t$.
11:      Observe $A_t = \Pi_{\mathrm{p}}^{\mathrm{up}}(\tilde{a}_{t+1}, \tau(t+1), V_t, \mathcal{H}_{t-1}^{\mathrm{up,p}}), X_{\tilde{a}_t, B_t}(t)$
12:      **if** $A_t = \tilde{a}_t$ **then** update history $\tilde{\mathcal{H}}_t^{\mathrm{down,p}}$.
13:      **end if**
14:      Update upstream player's history $\mathcal{H}_t^{\mathrm{up,p}}$.
15: **end for**

---

---

**Algorithm 2** Binary Search Subroutine

---

1: **Input:** action $a, N_T, \tilde{T}, \underline{\tau}_a, \overline{\tau}_a$.
2: **for** $d = 0, \ldots, N_T - 1$ **do**
3:      Compute $\tau_a^{\mathrm{mid}} = (\overline{\tau}_a(d) + \underline{\tau}_a(d))/2, T_a^{\neq} = 0$.
4:      **for** $t = d\tilde{T} + 1, \ldots, d\tilde{T} + \tilde{T}$ **do**
5:          Propose transfer $\tau_a^{\mathrm{mid}}(d)$ on arm $a$ and nothing for any other action $a' \in \mathcal{A}$.
6:          $A_t = \Pi_{\mathrm{p}}^{\mathrm{up}}(t, \tau^{\mathrm{mid}}(a), a, V_t, \mathcal{H}_{t-1}^{\mathrm{up,p}})$
7:          **if** $A_t \neq a$ **then**: $T^{\neq} + = 1$
8:          **end if**
9:          Update upstream player's history $\mathcal{H}_t^{\mathrm{up,p}}$.
10:      **end for**
11:      **if** $C\tilde{T}^{\kappa+\beta/\alpha} < T_a^{\neq} < \tilde{T} - C\tilde{T}^{\kappa+\beta/\alpha}$ **then** Return $\underline{\tau}_a(d), \overline{\tau}_a(d)$.
12:      **else if** $T_a^{\neq} \leqslant \tilde{T} - C\tilde{T}^{\kappa+\beta/\alpha}$ **then** $\overline{\tau}_a(d) = \tau_a^{\mathrm{mid}}(d) + 1/T^\beta$ and update history $\tilde{\mathcal{H}}_t^{\mathrm{down,p}}$.
13:      **else** $\underline{\tau}_a(d) = \tau_a^{\mathrm{mid}}(d) - 1/T^\beta$ and update history $\tilde{\mathcal{H}}_t^{\mathrm{down,p}}$.
14:      **end if**
15: **end for**

---

**Theorem 3.** *Assume that **H**1 and **H**2 hold. Then* BELGIC*, run with $\alpha, \beta$ satisfying (13) and any bandit subroutine* Bandit-Alg*, has an overall regret $\mathfrak{R}_{\mathrm{p}}^{\mathrm{down}}$ such that*

$$\mathfrak{R}_{\mathrm{p}}^{\mathrm{down}}(T, \Pi_{\mathrm{p}}^{\mathrm{up}}, \text{BELGIC}) \leqslant 2(3 + 2C + \bar{v} - \underline{v})\log_2(T)(2T^{1-\alpha\zeta} + T^{(\kappa+1)/2} + \lceil T^\alpha \rceil) + 4T^{1-\beta}$$
$$+ \mathfrak{R}_{\text{Bandit-Alg}}(T, \nu, \{\hat{\tau}_a\}_{a \in \mathcal{A}})$$

*where, for ease of notation*

$$\bar{v} = \max_{a,b \in \mathcal{A} \times \mathcal{A}} \{v^{\mathrm{down}}(a,b)\} \quad and \quad \underline{v} = \min_{a,b \in \mathcal{A} \times \mathcal{A}} \{v^{\mathrm{down}}(a,b)\}.$$

**Knowledge of** C **and** $\kappa$**.** An upper bound on C and $\kappa$ is sufficient to compute the hyperparameters in BELGIC. Theorem 3 shows that the bigger C and $\kappa$ are, the worse is the downstream player's regret, hence the interest of knowing them more precisely.

**Corollary 1.** *Assume that the upstream player's distribution $(\gamma_a)_{a \in \mathcal{A}}$ is such that **H** 1 holds. In addition, suppose that the distributions $(\gamma_a)_{a \in \mathcal{A}}$ and $(\nu_{a,b})_{a,b \in \mathcal{A} \times \mathcal{A}}$ are 1-sub-Gaussian and that the upstream player plays $\Pi_{\mathrm{p}}^{\mathrm{up}} = Algorithm$ 3 (a slight modification of UCB to take into account the incentives). Then the downstream player's regret when she runs* BELGIC *with parameters $\alpha = 3/4$ and $\beta = 1/4$ (which satisfy (13)) and subroutine* Bandit-Alg = UCB *satisfies the following upper*

*bound*[3]

$$\mathfrak{R}_{\mathrm{p}}^{\mathrm{down}}(T, \mathit{UCB}, \mathtt{BELGIC}) \leqslant \left(10 + 4K + 32\sqrt{K \log_2(KT^3)} + \bar{v} - \underline{v}\right) \log_2(T)(3 + 2T^{3/4})$$
$$+ 3K^2(\bar{v} - \underline{v}) \,.$$

The upper bound on the social welfare regret in Lemma 1 together with Corollary 1 shows that when the upstream player runs $\Pi_{\mathrm{p}}^{\mathrm{up}} = \mathtt{UCB}$ and the downstream player runs $\mathtt{BELGIC}$, the social welfare regret then satisfies $\mathfrak{R}^{\mathrm{sw}}(T, \Pi_{\mathrm{p}}^{\mathrm{up}}, \mathtt{BELGIC}) = \mathcal{O}(K \log(T) T^{(\kappa+1)/2})$.

In other words, if the downstream player runs $\mathtt{BELGIC}$ which produces a policy $\Pi_{\mathrm{down}}^{\mathrm{n}}$, for any upstream policy $\Pi_{\mathrm{p}}^{\mathrm{up}}$, $(\Pi_{\mathrm{down}}^{\mathrm{n}}, \Pi_{\mathrm{p}}^{\mathrm{up}})$ is welfare efficient.

**Influence of the upstream performance.** It is interesting to note that in the downstream player's regret bound, the upstream player's regret bound in $\mathcal{O}(T^\kappa)$ plays a significant role: the downstream player never learns faster than the upstream player. The latter's performance determines the social welfare convergence rate towards the social optimum. We can observe that the players' bounded rationality [Selten, 1990, Jones, 1999] and personal interest make the game converge towards the optimal social welfare equilibrium—even though they are both learning here.

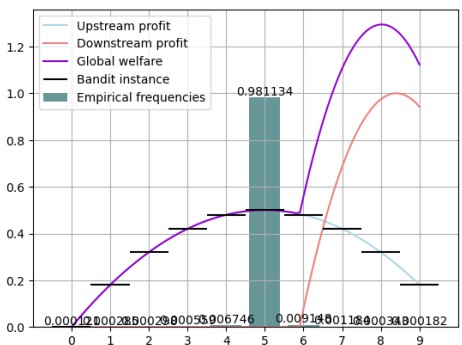 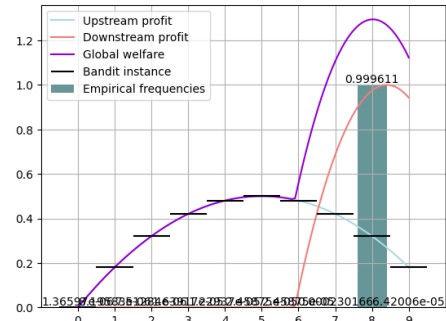

Figure 1: Empirical frequencies of the upstream player's actions when property rights are not defined (left) and when they are defined (right).

**Experiments.** We conclude this section with experiments showing the empirical convergence of our algorithm to a social optimum. In the simulation, we consider two firms, with firm 1 being upstream and firm 2 being downstream. Their profit functions are respectively given by

$$\pi_1 : \mapsto \max\left\{ q_1 - 2\left(\frac{q_1}{10}\right)^2 + 2\frac{q_1}{10}, 0 \right\},$$

and

$$\pi_2(q_1, q_2) \mapsto \max\left\{ -16\left(\frac{q_1}{10} - \frac{6}{10}\right)^2 + 8\left(q_1 - \frac{6}{10}\right)^2 + \frac{1}{50}q_2, 0 \right\}.$$

Thus, firm 1's and firm 2's profit functions depends quadratically on $q_1$ with an firm 1 optimum at $q_1' = 5$ and a social optimum at $q_1^\star = 8$. Note that in the expression of $\pi_2$, $q_2$ has very little influence as compared to $q_1$ - which allows to plot profits for only one value of $q_2$.

We discretize the setup, consider a bandit instance (horizon $T = 5.10^6$, 10 arms, average over 10 rounds) and we assume that $\mathtt{UCB}$ is used as a subroutine. In the first setting, there are no property rights and each firm runs $\mathtt{UCB}$ on their side. Second, property rights are defined and firm 2 runs $\mathtt{BELGIC}$ as its policy. The plots in Figure 1 display the empirical frequencies and show empirically the effectiveness of $\mathtt{BELGIC}$ to mitigate externalities.

## 4 Related work

Our work addresses the impact of externalities and is therefore related to taxation theory [see, e.g., Mirrlees et al., 2011, Salanie, 2011, and the references therein], a prominent solution for this issue, as

---

[3]Note that it is $K$ and not $\sqrt{K}$ here, since the action space is of cardinality $K^2$ for the downstream player.

exemplified by the *Pigouvian tax* [see Pigou, 2017]. Taxation is a fundamental aspect of all developed economies, with $30\%$ to $50\%$ of national income derived from taxes. The topic has been fruitful for various scenarios, including the carbon tax [Carattini et al., 2018, Metcalf and Weisbach, 2009], alcohol markets [Griffith et al., 2019], or business taxation [Boadway and Bruce, 1984]. Taxation can also be studied through an operations research lens, where it is used to enhance system efficiency or manage specific games [Roughgarden, 2010, Caragiannis et al., 2010, Bilò and Vinci, 2019]. Recent work by Cui et al. [2024] explores online mechanisms to maximize efficiency in congestion games.

Mechanism designs [Myerson, 1989, Nisan and Ronen, 1999, Laffont and Martimort, 2009] allow to design games that have specific desired outcomes. Deploying these mechanisms in their classical ecnomical form assume that players' utility functions are known a priori, which is often unrealistic. There is a major need to blend mechanism design with machine learning.

However, our approach differs, since, drawing inspiration from Coase's theory, we implement an online version of his theorem [Coase, 2013, Cooter, 1982], incorporating uncertainty to tackle the breakdown of social welfare in an online setting. We use the bandit setup [see Lattimore and Szepesvári, 2020, Slivkins et al., 2019] as a general and convenient way to model the game introduced by Coase. However, our work differs from considering a single agent playing a bandit game. Instead, we focus on the more general problem of multi-players bandits, a field receiving a growing attention from the community [see e.g., Boursier and Perchet, 2019, 2022, Sankararaman et al., 2019].

Our approach is inspired by the principal-agent model introduced by Dogan et al. [2023b], which was further extended by Dogan et al. [2023a], Scheid et al. [2024]. However, unlike the models proposed in the work of Dogan et al. [2023b,a], we do not specify a particular bandit algorithm for the upstream player and instead, we allow him to use any no-regret algorithm satisfying **H**2 in a black-box fashion. Conversely, the model of Scheid et al. [2024] assumes that the upstream player is always fully informed and best-responding, whereas we assume that he is also learning. Chen et al. [2023b] leverages a similar model to study information acquisition by a principal through an agent's actions. However, in their model, the agent is also almighty and knows exactly the costs associated with each action. Designing incentives in an unknown environment is related to auction theory incorporating uncertainty, as it is explored in the work of Feng et al. [see 2018], Li et al. [see 2023]. Similar issues have been explored in the *Reinforcement Learning* framework within a leader-follower game [see Chen et al., 2023a, with quantal responses by the follower] or in a principal-agent game with incentive design as done by Ben-Porat et al. [2023]. Donahue et al. [2024] also study a two-players repeated Stackelberg game on a bandit instance but instead of allowing for transfers, their main focus concerns the achievability of a *Stackelberg equilibrium* through iterations of bandit policies: the same kind of goal also appears in Collina et al. [2023]. Such principal-agent setups are of some interest to model various real-world situations such as the design of fundings for hospitals [Wang et al., 2024] or have been studied with multiple agents through the lens of auction design in dynamic setups [Bergemann and Said, 2010, Chen et al., 2023c], or to account for fairness [Fallah et al., 2024].

In our game, the downstream player needs to learn the optimal transfers/incentives to offer to the upstream player. This is related to the *Incentivized Exploration* literature [Mansour et al., 2016, Simchowitz and Slivkins, 2023, Esmaeili et al., 2023], which is often cast in terms of a benevolent planner who aims to optimize the global welfare of agents via plausible recommendations. A related model is *Bayesian Persuasion* [Kamenica and Gentzkow, 2011], where a sender influences a receiver's action through sending a signal. This model has begun to be studied in learning settings [see, e.g., Castiglioni et al., 2020, Bernasconi et al., 2022, Wu et al., 2022b,a].

# 5 Conclusion

This paper studies a model of externalities in a two-players sequential game where both players learn their optimal actions. We first show that when the players act independently, then a misalignment between the players' interests and the social welfare leads to a breakdown of the global utility. We then introduce interactions through transfers, which restores a social welfare optimum, representing the online version of the *Coase theorem*. To that purpose, we propose a policy for the downstream player which allows her to estimate the optimal transfers as well as choosing the best actions. The mathematical difficulty comes from the learning aspect on both sides. Since our work is coined in a learning framework for mechanism design, several directions for research are open, as for instance extensions to the multi-agent setting, which raises many questions.

## Acknowledgements

Funded by the European Union (ERC, Ocean, 101071601). Views and opinions expressed are however those of the author(s) only and do not necessarily reflect those of the European Union or the European Research Council Executive Agency. Neither the European Union nor the granting authority can be held responsible for them.

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

# A  Algorithmic Subroutine for the Binary Search

**Optimal Transfer.** For any given round $t \geqslant 1$, action $a \in \mathcal{A}$ and $\varepsilon > 0$, the downstream player can incentivize any best-responding upstream player to choose $a$ by offering a transfer, $\tau_a^{\star,\varepsilon} \in \mathbb{R}_+$, defined as:

$$\tau_a^{\star,\varepsilon} = \max_{a' \in \mathcal{A}} v^{\mathrm{up}}(a') - v^{\mathrm{up}}(a) + \varepsilon \ .$$

With this transfer, it holds that for any $a' \in \mathcal{A}, a' \neq a$, we have $v^{\mathrm{up}}(a') < v^{\mathrm{up}}(a) + \tau_a^{\star,\varepsilon}$, ensuring the upstream player's action $A_t = a$, since action $a$ yields a superior reward. Consequently,

$$\tau_a^{\star} = \lim_{\varepsilon \to 0} \tau_a^{\star,\varepsilon} = \max_{a' \in \mathcal{A}} v^{\mathrm{up}}(a') - v^{\mathrm{up}}(a)$$

represents the infimal transfer necessary to make arm $a$ the best upstream player's choice.

**First step of** `BELGIC`**: estimation of the optimal transfers.** Suppose that we consider an arm $a \in \mathcal{A}$ and that the downstream player offers an incentive $\tau_a$ to the upstream player if he picks this arm. We consider this procedure with a constant incentive $\tau_a$ for a batch of time steps of length $\tilde{T} = \lceil T^\alpha \rceil$ due to the fact that the upstream player is learning [see Perchet et al., 2016, for batched bandits in the usual multi-armed setting]. Lemma 2 shows that for `BELGIC` to accurately estimate $\tau_a^{\star}$ with high probability, it must hold

$$\mathrm{C}\tilde{T}^{\kappa+\beta/\alpha} < \tilde{T}/2 \ , \ \text{ which is equivalent to } \ \mathrm{C}T^{\kappa\alpha+\beta-\alpha} < 1/2 \ , \tag{16}$$

This is why we impose the condition (13) on $\alpha$ and $\beta$, namely $\beta/\alpha < 1 - \kappa$, which ensures that $\kappa(\alpha - 1) + \beta < 0$ and therefore $\lim_{T \to +\infty} T^{\kappa(\alpha-1)+\beta} = 0$. More precisely, we define for $a \in [K]$

$$\Lambda_a = (a - 1)\lceil \log_2 T^\beta \rceil \tilde{T} \ ,$$

which is the step after which starts the binary search procedure on arm $a$. For any $d \in \{1, \ldots, \lceil \log_2 T^\beta \rceil\}$ on arm $a$, we define

$$k_{a,d} = \Lambda_a + (d - 1)\tilde{T} \tag{17}$$

as the step after which starts the $d$-th batch iteration on arm $a$, and we consider

$$T_{a,d}^{\neq} = \mathrm{Card} \left\{ t \in \{k_{d,a} + 1, \ldots, k_{d,a} + \tilde{T}\} \text{ such that } A_t \neq a \right\} \ , \tag{18}$$

where $(A_t)_{t \in \{1, \ldots, K\lceil \log_2 T^\beta \rceil \tilde{T}\}}$ is given by Algorithm 2. Lemma 2 shows that for any $a \in \mathcal{A}, d \in \{1, \ldots, \lceil \log_2 T^\beta \rceil\}$, with high probability

$$\text{if } T_{d,a}^{\neq} < \tilde{T} - \mathrm{C}\tilde{T}^{\kappa+\beta/\alpha} \text{ and } T_{d,a}^{\neq} > \mathrm{C}\tilde{T}^{\kappa+\beta/\alpha}, \text{ then } |\tau_a^{\star} - \hat{\tau}_{d,a}| \leqslant 1/T^\beta \ , \tag{19}$$

where $\hat{\tau}_{a,d}$ is the current estimate of $\tau_a^{\star}$ offered for iteration $k_{a,d}$. In case (18) does not hold, it means that the upstream player has misplayed and has chosen most of the steps a suboptimal action, leading to an instantaneous regret for him larger than the bound given in **H**2. This is why (19) holds with high probability. To sum up, the first phase of `BELGIC` consists in $\lceil \log_2 T^\beta \rceil$ batches of binary search on each arm $a \in \mathcal{A}$ to obtain a precision level $1/T^\beta$ on the optimal transfer $\tau_a^{\star}$.

During this phase in Algorithm 2, we define $\overline{\tau}_a(d) \in \mathbb{R}_+$ as the upper estimate and $\underline{\tau}_a(d) \in \mathbb{R}_+$ as the lower estimate of $\tau_a^{\star}$ after $d \in \{1, \ldots, \lceil \log_2 T^\beta \rceil\}$ rounds of binary search on arm $a$. For any $t \in [T]$ and $a \in \mathcal{A}$, we define $\tau_a^{\mathrm{mid}}(t) = (\overline{\tau}_a(t) + \underline{\tau}_a(t))/2$. $\overline{\tau}_a(d), \underline{\tau}_a(d), \tau_a^{\mathrm{mid}}(d)$ are updated at the end of the $d$-th binary search batch of length $\tilde{T}$ on arm $a$. We define $N_T = \lceil \log_2 T^\beta \rceil$ as the number of binary search steps per arm.

After this first binary search phase, the downstream player computes estimates of the optimal transfers $\tau_a^{\star}$

$$(\hat{\tau}_a)_{a \in \mathcal{A}} = (\overline{\tau}_a(\lceil \log_2 T^\beta \rceil) + 1/T^\beta + \mathrm{C}T^{(\kappa-1)/2})_{a \in \mathcal{A}} \ ,$$

and offers these transfers $(\tau_a^{\star})_{a \in \mathcal{A}}$ to make the upstream player play any action $\tilde{a} \in \mathcal{A}$ she wants.

**Second Step.** After the first phase during which the optimal incentives are estimated by the downstream player through $(\hat{\tau}_a)_{a \in \mathcal{A}}$, she runs in the second phase the subroutine `Bandit-Alg` on the $\mathcal{A} \times \mathcal{A}$ bandit instance driven by her action $B_t$ and the upstream player's one $A_t$. More precisely, any bandit subroutine `Bandit-Alg`, such as `UCB` or $\varepsilon$-`greedy`, for instance, can be run in a black-box fashion

on the shifted bandit instance, where rewards are shifted by the upper estimated transfers $(\hat{\tau}_a)_{a \in \mathcal{A}}$. The downstream player computes a shifted history $\tilde{\mathcal{H}}_t^{\text{down,p}} = \varnothing$ for any $t \leqslant K \lceil T^\alpha \rceil \lceil \log_2 T^\beta \rceil$ and for any $t > K \lceil T^\alpha \rceil \lceil \log_2 T^\beta \rceil$

$$
\tilde{\mathcal{H}}_t^{\text{down,p}} = \left\{ \begin{array}{ll} \{\tilde{a}_t, B_t, \tau(t), U_t, X_{\tilde{a}_t, B_t}(t) - \hat{\tau}_{\tilde{a}_t}\} \cup \tilde{\mathcal{H}}_{t-1}^{\text{down,p}} & \text{if } \tilde{a}_t = A_t \\ \tilde{\mathcal{H}}_{t-1}^{\text{down,p}} & \text{otherwise} . \end{array} \right.
$$

which serves to feed `Bandit-Alg`, following `Bandit-Alg`: $(U_t, \tilde{\mathcal{H}}_{t-1}^{\text{down,p}}) \mapsto (\tilde{a}_t, B_t) \in \mathcal{A} \times \mathcal{A}$. Note that here, `Bandit-Alg` aims to maximize the shifted reward $(\nu^{\text{down}}(a, b) - \hat{\tau}_a)_{(a,b) \in \mathcal{A}^2}$.

Based on this decision by `Bandit-Alg`, $\Pi_{\text{p}}^{\text{down}}$ offers the incentive $\hat{\tau}_{\tilde{a}_t}$ associated with action $\tilde{a}_t$ and plays action $B_t$. Lemma 5 ensures that $\tilde{a}_t$ is the upstream player's best choice. Therefore, **H**2 ensures that the upstream player will not deviate from the downstream player's recommendation with high probability.

## B  Invariance when the property rights are given to the downstream player

Our focus in the paper was the case where the upstream player possesses the bandit instance and receives monetary payments. We argue here that the symmetric situation, i.e. when the property rights are given to the downstream player, can be analysed in the exact same way.

Assume that the the downstream player owns the bandit instance. This implies that (i) they can prescribe what arm the upstream player has to play at each round, and (ii) the upstream player may perform a monetary transfer to influence the arm they are allowed to pull.

Consider the same bandit setup as before. Formally, the downstream player's action is $(A_t, B_t) \in \mathcal{A} \times \mathcal{A}$ where $A_t$ is the arm that the upstream player is prescribed to pull (he cannot deviate since the downstream player has the property rights), while $B_t$ is the arm played by the downstream player. On the other hand, the upstream player's policy outputs at each round the action $(\tilde{a}_t, \tau(t)) \in \mathcal{A} \times \mathbb{R}_+$, where $\tilde{a}_t$ is the arm they choose to incentivize and $\tau(t)$ is the amount of transfer. It means that the downstream player receives a transfer $\tau(t)$ if she prescribes action $A_t = \tilde{a}_t$. As a consequence, the instantaneous utility of the upstream player is $Z_{A_t} - \mathbb{1}_{\tilde{a}_t}(A_t)\tau(t)$, while the downstream player receives $X_{A_t, B_t} + \mathbb{1}_{\tilde{a}_t}(A_t)\tau(t)$. In that case, the upstream player may perform a binary search on each arm $\bar{a} \in \mathcal{A}$ to identify the optimal incentive $\tau_{\bar{a}}^\star$, by considering Card $\{t \in [T] \text{ such that } \tilde{a}_t = \bar{a}\}$ during batches designed for the binary search and then play on the shifted bandit instance as we explained. This situation and the upstream player's strategy are now equivalent to the one presented before.

## C  Proofs and Technical Results

Recall that we defined the shifted history $\tilde{\mathcal{H}}_t^{\text{down,p}}$ that will serve to feed $\Pi_{\text{p}}^{\text{down}}$ at time $t$ as $\tilde{\mathcal{H}}_t^{\text{down,p}} = (\tilde{a}_s, \tau(s), A_s, V_s, Z_{A_s}(s))_{s \leqslant t}$.

**Theorem 4.** *Suppose that* $\operatorname{argmax}_{a \in \mathcal{A}} v^{\text{up}}(a)$ *is the singleton* $\{a_\star^{\text{u}}\}$ *and that*

$$
v^{\text{up}}(a^{\text{sw}}) + v^{\text{down}}(a^{\text{sw}}, b^{\text{sw}}) - v^{\text{up}}(a_\star^{\text{u}}) + v^{\text{down}}(a_\star^{\text{u}}, b) > 0 ,
$$

*for any* $b \in \mathcal{A}$. *In the absence of property rights and when the upstream player runs any no-regret policy* $\Pi_{\text{n}}^{\text{up}}$, *we have* $\mathfrak{R}^{\text{sw}}(T, \Pi_{\text{n}}^{\text{up}}, \Pi_{\text{n}}^{\text{down}}) \geqslant T\Delta^{\text{sw}} - \mathfrak{R}_{\text{n}}^{\text{up}}(T, \Pi_{\text{n}}^{\text{up}})\Delta^{\text{sw}}/\Delta^{\text{up}}$, *where* $\Delta^{\text{up}} = \min_{a' \in \mathcal{A} \setminus \{a_\star^{\text{u}}\}} v^{\text{up}}(a_\star^{\text{u}}) - v^{\text{up}}(a')$ *and* $\Delta^{\text{sw}} = v^{\text{up}}(a^{\text{sw}}) + v^{\text{down}}(a^{\text{sw}}, b^{\text{sw}}) - \max_{b \in \mathcal{A}}(v^{\text{up}}(a_\star^{\text{u}}) + v^{\text{down}}(a_\star^{\text{u}}, b))$. *Therefore,* $\mathfrak{R}^{\text{sw}}(T, \Pi_{\text{n}}^{\text{up}}, \Pi_{\text{n}}^{\text{down}}) = \Omega(T)$ *and* $(\Pi_{\text{n}}^{\text{up}}, \Pi_{\text{n}}^{\text{down}})$ *is not welfare efficient.*

*Proof of Theorem 4.* Since $\operatorname{argmax}_{a \in \mathcal{A}} v^{\text{up}}(a)$ is the singleton $\{a_\star^{\text{up}}\}$, we define $\Delta^{\text{up}} = \min_{a' \in \mathcal{A} \setminus \{a_\star^{\text{u}}\}} v^{\text{up}}(a_\star^{\text{u}}) - v^{\text{up}}(a')$ as the upstream player reward gap and $\Delta^{\text{sw}} = v^{\text{up}}(a^{\text{sw}}) + v^{\text{down}}(a^{\text{sw}}, b^{\text{sw}}) - \max_{b \in \mathcal{A}}(v^{\text{up}}(a_\star^{\text{u}}) + v^{\text{down}}(a_\star^{\text{u}}, b))$ as the *social welfare* reward gap if the upstream player plays his most preferred action.

Denote $N_\star^{\text{up}}(T)$ the number of pulls of the upstream player up to time $T$ on the arm $a_\star^{\text{u}}$. By definition of $\Delta^{\text{up}}$, we have that for any step $t \in [T]$ such that $A_t \neq a_\star^{\text{up}}$, $\max_{a \in \mathcal{A}}\{v^{\text{up}}(a)\} - v^{\text{up}}(A_t) =$

$v^{\mathrm{up}}(a_\star^{\mathrm{up}}) - v^{\mathrm{up}}(A_t) \geqslant \min_{a' \in \mathcal{A} \setminus \{a_\star^{\mathrm{u}}\}} v^{\mathrm{up}}(a_\star^{\mathrm{u}}) - v^{\mathrm{up}}(a') = \Delta^{\mathrm{up}}$. There are $T - N_\star^{\mathrm{up}}(T)$ such steps, which leads to

$$\mathfrak{R}_{\mathrm{n}}^{\mathrm{up}}(T, \Pi_{\mathrm{n}}^{\mathrm{up}}, \Pi_{\mathrm{p}}^{\mathrm{down}}) \geqslant \mathbb{E}\left[\sum_{t=1}^{T} \max_{a \in \mathcal{A}}\{v^{\mathrm{up}}(a)\} - v^{\mathrm{up}}(A_t)\right]$$
$$\geqslant (T - \mathbb{E}[N_\star^{\mathrm{u}}(T)])\Delta^{\mathrm{up}} \,,$$

and we obtain

$$\mathbb{E}[N_\star^{\mathrm{u}}(T)] \geqslant T - \mathfrak{R}_{\mathrm{n}}^{\mathrm{up}}(T, \Pi_{\mathrm{n}}^{\mathrm{up}})/\Delta^{\mathrm{up}} \,.$$

Moreover, for any $t \in [T]$ such that $A_t = a_\star^{\mathrm{up}}$ and any $B_t \in \mathcal{A}$, $v^{\mathrm{up}}(a^{\mathrm{sw}}) + v^{\mathrm{down}}(a^{\mathrm{sw}}, b^{\mathrm{sw}}) - (v^{\mathrm{up}}(A_t) + v^{\mathrm{down}}(A_t, B_t)) = v^{\mathrm{up}}(a^{\mathrm{sw}}) + v^{\mathrm{down}}(a^{\mathrm{sw}}, b^{\mathrm{sw}}) - (v^{\mathrm{up}}(a_\star^{\mathrm{up}}) + v^{\mathrm{down}}(a^{\mathrm{up}}, B_t)) \geqslant \Delta^{\mathrm{sw}}$ by definition, which leads to

$$\mathfrak{R}^{\mathrm{sw}}(T, \Pi_{\mathrm{n}}^{\mathrm{up}}, \Pi_{\mathrm{n}}^{\mathrm{down}}) \geqslant \mathbb{E}[N_\star^{\mathrm{u}}(T)]\Delta^{\mathrm{sw}} \,,$$

and we obtain

$$\mathfrak{R}^{\mathrm{sw}}(T, \Pi_{\mathrm{n}}^{\mathrm{up}}, \Pi_{\mathrm{n}}^{\mathrm{down}}) \geqslant T\Delta^{\mathrm{sw}} - \mathfrak{R}_{\mathrm{n}}^{\mathrm{up}}(T, \Pi_{\mathrm{n}}^{\mathrm{up}})\Delta^{\mathrm{sw}}/\Delta^{\mathrm{up}} \,.$$

Since $\lim_{T \to +\infty} \mathfrak{R}_{\mathrm{n}}^{\mathrm{up}}(T, \Pi_{\mathrm{n}}^{\mathrm{up}})/T = 0$, we have $\lim_{T \to +\infty} \mathfrak{R}^{\mathrm{sw}}(T, \Pi_{\mathrm{n}}^{\mathrm{up}}, \Pi_{\mathrm{n}}^{\mathrm{down}})/T = \Delta^{\mathrm{sw}} > 0$, hence the result. $\qquad\square$

The proof of Theorem 2 is an immediate consequence of Theorem 4.

**Lemma 1.** *Recall that $\mu^{\star,\mathrm{down}}$ is the downstream player's optimal reward as defined as a solution of* (10). *We have $\mu^{\star,\mathrm{down}} = \max_{a,b \in \mathcal{A}}\{v^{\mathrm{down}}(a,b) + v^{\mathrm{up}}(a)\} - \max_{a' \in \mathcal{A}}\{v^{\mathrm{up}}(a')\}$, as well as $(a^{\mathrm{opt}}, b^{\mathrm{opt}}) = (a^{\mathrm{sw}}, b^{\mathrm{sw}})$ and $\mu^{\star,\mathrm{up}} + \mu^{\star,\mathrm{down}} = v^{\mathrm{up}}(a^{\mathrm{sw}}) + v^{\mathrm{down}}(a^{\mathrm{sw}}, b^{\mathrm{sw}}) = \max_{a,b \in \mathcal{A}}\{v^{\mathrm{up}}(a) + v^{\mathrm{down}}(a,b)\}$, where $\mu^{\star,\mathrm{up}}$ is defined in Equation* (1). *Moreover, for any integer $T \in \mathbb{N}^\star$, and policies $\Pi_{\mathrm{p}}^{\mathrm{up}}, \Pi_{\mathrm{p}}^{\mathrm{down}}$, we have that*

$$\mathfrak{R}^{\mathrm{sw}}(T, \Pi_{\mathrm{p}}^{\mathrm{up}}, \Pi_{\mathrm{p}}^{\mathrm{down}}) \leqslant \mathfrak{R}_{\mathrm{p}}^{\mathrm{up}}(T, \Pi_{\mathrm{p}}^{\mathrm{up}}, \Pi_{\mathrm{p}}^{\mathrm{down}}) + \mathfrak{R}_{\mathrm{p}}^{\mathrm{down}}(T, \Pi_{\mathrm{p}}^{\mathrm{up}}, \Pi_{\mathrm{p}}^{\mathrm{down}}) \,.$$

*Proof of Lemma 1.* Recall that $\mu^{\star,\mathrm{down}}$ is defined as $\mu^{\star,\mathrm{down}} = \sup_{a,b \in \mathcal{A}^2, \tau \in \mathbb{R}_+}\{v^{\mathrm{down}}(a,b) - \tau\}$, such that $a \in \operatorname{argmax}_{a' \in \mathcal{A}}\{v^{\mathrm{up}}(a') + \mathbb{1}_a(a')\tau\}$ and $a_\star^{\mathrm{up}} = \operatorname{argmax}_{a' \in \mathcal{A}} v^{\mathrm{up}}(a)$. Note that we can write

$$\mu^{\star,\mathrm{down}} = \max\{\sup_{a,b \in \mathcal{A}^2, \tau \in \mathbb{R}_+} \mathbb{1}_{\tilde{\mathsf{A}}}(a,\tau)(v^{\mathrm{down}}(a,b) - \tau), \max_{b \in \mathcal{A}} v^{\mathrm{down}}(a_\star^{\mathrm{up}}, b)\} \,,$$

where $\tilde{\mathsf{A}} = \{(a,\tau) \colon v^{\mathrm{up}}(a) + \tau \geqslant \max_{a'} v^{\mathrm{up}}(a') + \mathbb{1}_a(a')\tau\}$ which is the set of pairs $(a,\tau) \in \mathcal{A} \times \mathbb{R}_+$ such that the constraint binds. However, we also have by definition that $v^{\mathrm{up}}(a_\star^{\mathrm{up}}) + 0 \geqslant \max_{a' \in \mathcal{A}} v^{\mathrm{up}}(a') + \mathbb{1}_{a_\star^{\mathrm{up}}}(a') \cdot 0$ and hence, $(a_\star^{\mathrm{up}}, 0) \in \tilde{\mathsf{A}}$. Therefore, since $v^{\mathrm{down}}(a_\star^{\mathrm{up}}, b) \geqslant 0$ for any $b \in \mathcal{A}$, we can write

$$\mu^{\star,\mathrm{down}} = \sup_{(a,\tau) \in \tilde{\mathsf{A}}, b \in \mathcal{A}} \{v^{\mathrm{down}}(a,b) - \tau\} \,.$$

First note that if $(a,\tau) \in \tilde{\mathsf{A}}$, then for any $a' \in \mathcal{A}, \tau \in \mathbb{R}_+$, $v^{\mathrm{up}}(a) + \tau \geqslant v^{\mathrm{up}}(a') + \mathbb{1}_a(a')\tau$, which gives

$$v^{\mathrm{up}}(a) - v^{\mathrm{up}}(a') \geqslant (\mathbb{1}_a(a') - 1)\tau \,.$$

However, either $a = a'$ and hence $v^{\mathrm{up}}(a) - v^{\mathrm{up}}(a') = 0$, either $a \neq a'$ and hence $\mathbb{1}_a(a') = 0$. Therefore, we have that

$$v^{\mathrm{up}}(a) - v^{\mathrm{up}}(a') \geqslant -\tau \,,$$

which implies by definition of the optimal incentives that $\tau \geqslant v^{\mathrm{up}}(a') - v^{\mathrm{up}}(a)$ for any $a' \in \mathcal{A}$, and hence $\tau_a^\star \leqslant \tau$ for any $(a,\tau) \in \tilde{\mathsf{A}}$.

In addition, $(a, \tau_a^\star) \in \tilde{\mathsf{A}}$ by definition. Consequently,

$$\mu^{\star,\mathrm{down}} = \max_{a,b \in \mathcal{A}^2} \{v^{\mathrm{down}}(a,b) - \tau_a^\star\} = \max_{a,b \in \mathcal{A}} \{v^{\mathrm{down}}(a,b) - \max_{a' \in \mathcal{A}}\{v^{\mathrm{up}}(a')\} + v^{\mathrm{up}}(a)\} \,,$$

hence the first part of the result. Since $\mu^{\star,\mathrm{up}}$ is defined as $\mu^{\star,\mathrm{up}} = \max_{a\in\mathcal{A}} v^{\mathrm{up}}(a)$, we have that

$$
\begin{aligned}
\mu^{\star,\mathrm{down}} + \mu^{\star,\mathrm{up}} &= \max_{a,b\in\mathcal{A}}\{v^{\mathrm{down}}(a,b) - \max_{a'\in\mathcal{A}}\{v^{\mathrm{up}}(a')\} + v^{\mathrm{up}}(a)\} + \max_{a\in\mathcal{A}} v^{\mathrm{up}}(a) \\
&= \max_{a,b\in\mathcal{A}}\{v^{\mathrm{down}}(a,b) + v^{\mathrm{up}}(a)\} \\
&= v^{\mathrm{up}}(a^{\mathrm{sw}}) + v^{\mathrm{down}}(a^{\mathrm{sw}}, b^{\mathrm{sw}}) \ .
\end{aligned}
$$

Now summing $\mathfrak{R}_{\mathrm{p}}^{\mathrm{down}}$ and $\mathfrak{R}_{\mathrm{p}}^{\mathrm{up}}$ as defined in (9) and (11), we obtain

$$
\begin{aligned}
&\mathfrak{R}_{\mathrm{p}}^{\mathrm{up}}(T,\Pi_{\mathrm{p}}^{\mathrm{up}},\Pi_{\mathrm{p}}^{\mathrm{down}}) + \mathfrak{R}_{\mathrm{p}}^{\mathrm{down}}(T,\Pi_{\mathrm{p}}^{\mathrm{up}},\Pi_{\mathrm{p}}^{\mathrm{down}}) \\
&= \mathbb{E}\left[\sum_{t=1}^{T} \max_{a\in\mathcal{A}}\{v^{\mathrm{up}}(a) + \mathbb{1}_{\tilde{a}_t}(a)\tau(t)\} - (v^{\mathrm{up}}(A_t) + \mathbb{1}_{\tilde{a}_t}(A_t)\tau(t))\right] \\
&\quad + T\max_{a,b\in\mathcal{A}^2}\{v^{\mathrm{down}}(a,b) - \max_{a'\in\mathcal{A}}\{v^{\mathrm{up}}(a')\} + v^{\mathrm{up}}(a)\} - \mathbb{E}\left[\sum_{t=1}^{T} v^{\mathrm{down}}(A_t,B_t) - \mathbb{1}_{\tilde{a}_t}(A_t)\tau(t)\right] \\
&\geqslant T\max_{a\in\mathcal{A}}\{v^{\mathrm{up}}(a)\} - \mathbb{E}\left[\sum_{t=1}^{T} v^{\mathrm{up}}(A_t) + \mathbb{1}_{\tilde{a}_t}(A_t)\tau(t)\right] - \mathbb{E}\left[\sum_{t=1}^{T} v^{\mathrm{down}}(A_t,B_t) - \mathbb{1}_{\tilde{a}_t}(A_t)\tau(t)\right] \\
&\quad + T\max_{a,b\in\mathcal{A}^2}\{v^{\mathrm{down}}(a,b) + v^{\mathrm{up}}(a)\} - T\max_{a'\in\mathcal{A}}\{v^{\mathrm{up}}(a')\} \\
&= T\max_{a,b\in\mathcal{A}^2}\{v^{\mathrm{up}}(a) + v^{\mathrm{down}}(a,b)\} - \mathbb{E}\left[\sum_{t=1}^{T} v^{\mathrm{up}}(A_t) + v^{\mathrm{down}}(A_t,B_t)\right] \\
&= \mathfrak{R}^{\mathrm{sw}}(T,\Pi_{\mathrm{p}}^{\mathrm{up}},\Pi_{\mathrm{p}}^{\mathrm{down}}) \ ,
\end{aligned}
$$

hence the result. $\qquad\square$

For a downstream player's policy $\Pi_{\mathrm{p}}^{\mathrm{down}}$, we define $\tilde{\mathfrak{R}}_{\mathrm{p}}^{\mathrm{up}}$ as the upstream player's regret without expectation, following

$$
\tilde{\mathfrak{R}}_{\mathrm{p}}^{\mathrm{up}}(\{s+1,\ldots,s+t\},\Pi_{\mathrm{p}}^{\mathrm{up}},\Pi_{\mathrm{p}}^{\mathrm{down}}) = \sum_{l=s+1}^{s+t} \max_{a\in\mathcal{A}}\{v^{\mathrm{up}}(a) + \mathbb{1}_{\tilde{a}_l}(a)\tau(l)\} - (v^{\mathrm{up}}(A_l) + \mathbb{1}_{\tilde{a}_l}(A_l)\tau(l)) \ ,
$$

where $(\tilde{a}_l,\tau(l))_{l\in[T]}$ are the incentives output by $\Pi_{\mathrm{p}}^{\mathrm{down}}$ and $(A_l)_{l\in[T]}$ are the output of $\Pi_{\mathrm{p}}^{\mathrm{up}}$. Recall the assumption that we use on the upstream player's regret for a policy $\Pi_{\mathrm{p}}^{\mathrm{up}}$. We show here that it is satisfied by typical no-regret bandit algorithms.

**H2.** *There exist* $\mathrm{C}, \zeta > 0, \kappa \in [0,1)$ *such that for any* $s,t \in [T]$ *with* $s+t \leqslant T$, *any* $\{\tau_a\}_{a\in[K]} \in \mathbb{R}_+^K$ *and any policy* $\Pi_{\mathrm{p}}^{\mathrm{down}}$ *that offers almost surely a transfer* $(\tilde{a}_l,\tau(l)) = (\tilde{a}_l,\tau_{\tilde{a}_l})$ *for any* $l \in \{s+1,\ldots,s+t\}$, *the batched regret of the upstream player following* $\Pi_{\mathrm{p}}^{\mathrm{up}}$ *satisfies, with probability at least* $1 - t^{-\zeta}$,

$$
\sum_{l=s+1}^{s+t} \max_{a\in\mathcal{A}}\{v^{\mathrm{up}}(a) + \mathbb{1}_{\tilde{a}_l}(a)\tau_{\tilde{a}_l}\} - (v^{\mathrm{up}}(A_l) + \mathbb{1}_{\tilde{a}_l}(A_l)\tau_{\tilde{a}_l}) \leqslant \mathrm{C}t^{\kappa} \ .
$$

We present the upstream player's UCB subroutine.

**Proposition 2.** *Let* $s,t \in [T]$ *such that* $s+t \leqslant T$. *Suppose that there exists a family* $(\tau_a)_{a\in\mathcal{A}}$ *of constant incentives associated with each arm* $a \in \mathcal{A}$ *such that we have in Algorithm 3:* $(\tilde{a}_l,\tau(l))_{s+l\in\{s+1,\ldots,s+t\}} = (\tilde{a}_l,\tau_{\tilde{a}_l})_{l\in\{s+1,\ldots,s+t\}}$ *as an output of* $\Pi_{\mathrm{p}}^{\mathrm{down}}$. *Suppose that the distributions* $\gamma_a$ *are* 1-sub-Gaussian. *Then with probability at least* $1 - T^{-2}$, *the regret of the version of* UCB *given in Algorithm 3 run by the upstream player satisfies*

$$
\tilde{\mathfrak{R}}_{\mathrm{p}}^{\mathrm{up}}(\{s+1,\ldots,s+t\}, UCB, \Pi_{\mathrm{p}}^{\mathrm{down}}) \leqslant 8\sqrt{\log(KT^3)}\sqrt{tK} \ .
$$

Note that the major difference between this assumption and the regret bounds that we generally consider in multi-armed bandit problems is that we consider the regret without expectation here.

---

**Algorithm 3** Upstream player's UCB

---

1: **Input:** Set of arms $K$, horizon $T$.
2: **Initialize:** For any arm $a \in [K]$, set $\hat{\mu}_a = 0$, $T_a = 0$.
3: **for** $1 \leqslant t \leqslant K$: **do**
4:     Pull arm $A_t = t$
5:     Update $\hat{\mu}_{A_t} = X_{A_t}(t)$, $T_{A_t}(t) = 1$
6: **end for**
7: **for** $t \geqslant K + 1$ **do**
8:     Observe the incentive $(\tilde{a}_t, \tau(t))$.
9:     Pull arm $A_t \in \operatorname{argmax}_{a \in [K]} \left\{ \hat{\mu}_a(t-1) + 2\sqrt{\frac{\log(KT^3)}{T_a(t-1)}} + \mathbb{1}_{\tilde{a}_t}(a)\tau(t) \right\}$
10:     Update $T_{A_t}(t) = T_{A_t}(t-1) + 1$, $\hat{\mu}_{A_t}(t) = \frac{1}{T_{A_t}(t)}(T_{A_t}(t-1)\hat{\mu}_{A_t}(t-1) + X_{A_t}(t))$
11: **end for**

---

*Proof of Proposition 2.* The proof is adapted from the proof of Bubeck et al. [2012, Theorem 2.1]. Let $s, t \in [T]$ such that $s + t \leqslant T$. For any integer $l \in \{s+1, \ldots, s+t\}$, we write $n_l(a) = \text{Card}\{l' \in [l] \text{ such that } A_{l'} = a\}$ for the number of pulls of arm $a$ and $\hat{\mu}_a(l)$ for the empirical mean utility of the arm $a \in \mathcal{A}$ estimated on the batch $\{1, \ldots, l\}$: $\hat{\mu}_a(l) = n_l(a)^{-1} \sum_{k=1}^l \mathbb{1}_a(A_k)X_a(k)$. Since the rewards on the incentivized bandit instance are 1-sub-Gaussian, a Hoeffding bound gives that for any $\delta \in (0,1)$, $a \in \mathcal{A}$, $l \in \{s+1, \ldots, s+t\}$, and any family of arms $(a_{l'})_{l' \in \{1, \ldots, l\}}$ such that $\text{Card}\{j: a_j = a\} = k$, we have that

$$\mathbb{P}\Big( |\hat{\mu}_a(l) - v^{\mathrm{up}}(a)| \geqslant 2\sqrt{\log(2/\delta)/k} \,\Big|\, (A_{l'})_{l' \in \{1, \ldots, s+l\}} = (a_{l'})_{l' \in \{1, \ldots, s+l\}} \Big) \leqslant \delta \;.$$

Therefore, for any $\delta \in (0,1)$, $a \in \mathcal{A}$, $l \in \{s+1, \ldots, s+t\}$, we have the following bound

$$\mathbb{P}\Big( |\hat{\mu}_a(l) - v^{\mathrm{up}}(a)| \geqslant 2\sqrt{\log(2/\delta)/n_l(a)} \Big)$$

$$= \mathbb{P}\left( \bigcup_{k=1}^l \bigcup_{\substack{(a_{l'})_{l'} \text{ s.t.} \\ \text{Card}\{l' \in \{1, \ldots, l\}:\, a_{l'} = a\} = k}} \{|\hat{\mu}_a(l) - v^{\mathrm{up}}(a)| \geqslant 2\sqrt{\log(2/\delta)/k}\} \right)$$

$$\leqslant \sum_{k=1}^l \mathbb{P}\left( \bigcup_{\substack{(a_{l'})_{l'} \text{ s.t.} \\ \text{Card}\{l' \in \{1, \ldots, l\}:\, a_{l'} = a\} = k}} \{|\hat{\mu}_a(s+l) - v^{\mathrm{up}}(a)| \geqslant 2\sqrt{\log(2/\delta)/k}\} \right)$$

$$\leqslant \sum_{k=1}^l \sum_{\substack{(a_{l'})_{l'} \text{ s.t.} \\ \text{Card}\{l' \in \{1, \ldots, l\}:\, a_{l'} = a\} = k}} \mathbb{P}\Big( |\hat{\mu}_a(s+l) - v^{\mathrm{up}}(a)| \geqslant 2\sqrt{\log(2/\delta)/k} \,\Big|\, A_l = a_l \Big) \mathbb{P}(A_l = a_l)$$

$$\leqslant T\delta \;,$$

and with an union bound, we obtain that

$$\mathbb{P}\Big( \exists l \in [t], a \in \mathcal{A} \text{ such that } |\hat{\mu}_a(l) - v^{\mathrm{up}}(a)| \geqslant 2\sqrt{\log(2/\delta)/n_l(a)} \Big) \leqslant T^2 K \delta \;,$$

Considering the probability of the opposite event, we have that

$$\mathbb{P}\Big( \text{for any } l \in [t], a \in \mathcal{A}, |\hat{\mu}_a(l) - v^{\mathrm{up}}(a)| \leqslant 2\sqrt{\log(2T^2K/\delta)/n_l(a)} \Big) \geqslant 1 - \delta \;, \qquad (20)$$

where we rescaled $\delta$ as $\delta/T^2K$.

For the remaining of the proof, we take $\delta = T^{-2}$ and define $a_l^\star = \operatorname{argmax}_{a \in \mathcal{A}}\{v^{\mathrm{up}}(a) + \mathbb{1}_{\tilde{a}_l}(a)\tau\}$. We now assume that the event $\{$for any $l \in \{s+1, \ldots, s+t\}, a \in \mathcal{A}, |\hat{\mu}_a(l) - v^{\mathrm{up}}(a)| \leqslant 2\sqrt{\log(2T^2K/\delta)/n_l(a)}\}$ holds. If at some step $l \in \{s+1, \ldots, s+t\}$, action $A_l$ is chosen in Algorithm 3, it means that

$$\hat{\mu}_{A_l}(l) + 2\sqrt{\log(tK/\delta)/n_l(A_l)} + \mathbb{1}_{\tilde{a}_l}(A_l)\tau_{\tilde{a}_l} \geqslant \hat{\mu}_{a_l^\star}(l) + 2\sqrt{\log(tK/\delta)/n_l(a_l^\star)} + \mathbb{1}_{\tilde{a}_l}(a_l^\star)\tau_{\tilde{a}_l} \;,$$

with regards to the choice of actions in UCB based on the upper confidence bound. We now decompose the whole regret on the batch $\{s+1,\ldots,s+t\}$ defined as $\tilde{\mathfrak{R}}_{\mathrm{p}}^{\mathrm{up}}(\{s+1,\ldots,s+t\},\texttt{UCB},\Pi_{\mathrm{p}}^{\mathrm{down}})$. (20) ensures that with probability at least $1-1/T^2$

$$\tilde{\mathfrak{R}}_{\mathrm{p}}^{\mathrm{up}}(\{s+1,\ldots,s+t\},\texttt{UCB},\Pi_{\mathrm{p}}^{\mathrm{down}}) = \sum_{l=s+1}^{s+t} v^{\mathrm{up}}(a_l^\star) + \mathbb{1}_{\tilde{a}_l}(a_l^\star)\tau_{\tilde{a}_l} - (v^{\mathrm{up}}(A_l) + \mathbb{1}_{\tilde{a}_l}(A_l)\tau_{\tilde{a}_l})$$

$$\leqslant \sum_{l=s+1}^{s+t} \hat{\mu}_{a_l^\star}(l) + 2\sqrt{\log(Kt/\delta)/n_l(a_l^\star)} - v^{\mathrm{up}}(A_l) + \mathbb{1}_{\tilde{a}_l}(a_l^\star)\tau_{\tilde{a}_l} - \mathbb{1}_{\tilde{a}_l}(A_l)\tau_{\tilde{a}_l}$$

$$\leqslant \sum_{l=s+1}^{s+t} \hat{\mu}_{A_l}(l) + 2\sqrt{\log(Kt/\delta)/n_l(A_l)} - v^{\mathrm{up}}(A_l)$$

$$\leqslant \sum_{l=s+1}^{s+t} \hat{\mu}_{A_l}(l) + 2\sqrt{\log(Kt/\delta)/n_l(A_l)} - (\hat{\mu}_{A_l}(l) - 2\sqrt{\log(Kt/\delta)/n_l(A_l)})$$

$$\leqslant 4\sqrt{\log(Kt/\delta)} \sum_{l=s+1}^{s+t} \sqrt{1/n_l(A_l)} \,,$$

and we have

$$\sum_{l=s+1}^{s+t} \sqrt{1/n_l(A_l)} \leqslant \sum_{i=1}^{K} \sum_{l=s+1}^{s+t} \sqrt{\mathbb{1}_{A_l}(i)/n_l(i)}$$

$$\leqslant \sum_{i=1}^{K} \sum_{j=n_{s+1}(i)}^{n_{s+t}(i)} 1/\sqrt{j} \leqslant 2\sum_{i=1}^{K} \sqrt{n_{s+t}(i) - n_s(i)} \,, \tag{21}$$

where the last step holds because for any integers $s,t \in \mathbb{N}^\star$, we have that

$$\sum_{l=s+1}^{s+t} \frac{1}{\sqrt{l}} = \sum_{l=s+1}^{s+t} \frac{1}{\sqrt{l}} \int_{x=l-1}^{l} \mathrm{d}x \leqslant \sum_{l=s+1}^{s+t} \int_{x=l-1}^{l} \frac{\mathrm{d}x}{\sqrt{x}} = \int_{x=s}^{s+t} \frac{\mathrm{d}x}{\sqrt{x}} = 2(\sqrt{s+t} - \sqrt{s}) \,.$$

Using Cauchy-Schwarz inequality we obtain from (21)

$$1/K \sum_{l=s+1}^{s+t} \sqrt{1/n_l(A_l)} \leqslant 2\sqrt{1/K \sum_{i=1}^{K} n_{s+t}(i) - n_s(i)} = 2\sqrt{t/K} \,,$$

which gives $\sum_{l=s+1}^{s+t} \sqrt{1/n_l(A_l)} \leqslant 2\sqrt{tK}$.

Finally plugging all the terms together, since $\delta = 1/T^2$, we obtain that with probability at least $1-1/T^2$

$$\tilde{\mathfrak{R}}_{\mathrm{p}}^{\mathrm{up}}(\{s+1,\ldots,s+t\},\texttt{UCB}) \leqslant 8\sqrt{\log(KT^3)}\sqrt{tK} \,.$$

$\square$

**Lemma 2.** *Assume **H2** holds and consider some arm $a \in \mathcal{A}$ such that we run the $d$-th batch of binary search on $a$ with $d \in \{1,\ldots,\lceil \log_2 T^\beta \rceil\}$: for any $t \in \{k_{a,d}+1,\ldots,k_{a,d}+\tilde{T}\}$, we have $(\tilde{a}_t,\tau(t)) = (a,\tau_a)$ with $\tau_a = \tau_a^{\mathrm{mid}}(d)$ and $\tilde{T} = \lceil T^\alpha \rceil$. Recall that we defined in (18): $T_{a,d}^{\neq} = \mathrm{Card}\{t \in \{k_{a,d}+1,\ldots,k_{a,d}+\tilde{T}\} \text{ such that } A_t \neq a\}$. Let $\beta \in (0,1)$ be such that $\beta < \alpha(1-\kappa)$. Given that the event $\{\tilde{\mathfrak{R}}_{\mathrm{p}}^{\mathrm{up}}(\{k_{a,d}+1,\ldots,k_{a,d}+\tilde{T}\},\Pi_{\mathrm{p}}^{\mathrm{up}},\Pi_{\mathrm{p}}^{\mathrm{down}}) \leqslant \mathrm{C}\tilde{T}^\kappa\}$ holds, we have that*

- *If $T_{a,d}^{\neq} < \tilde{T} - \mathrm{C}\tilde{T}^{\kappa+\beta/\alpha}$, then $\tau_a^\star < \tau_a + 1/T^\beta$.*

- *If $T_{a,d}^{\neq} > \mathrm{C}\tilde{T}^{\kappa+\beta/\alpha}$, then $\tau_a^\star > \tau_a - 1/T^\beta$.*

*Consequently, with probability at least $1 - 2T^{-\alpha\zeta}$, if $\mathrm{C}\tilde{T}^{\kappa+\beta/\alpha} < T_{a,d}^{\neq} < \tilde{T} - \mathrm{C}\tilde{T}^{\kappa+\beta/\alpha}$, then $|\tau_a^\star - \tau_a| \leqslant 1/T^\beta$.*

*Proof of Lemma 2.* The whole proof is done conditionally on the event

$$\{\tilde{\mathfrak{R}}_{\mathrm{p}}^{\mathrm{up}}(\{k_{a,d}+1,\ldots,k_{a,d}+\tilde{T}\},\Pi_{\mathrm{p}}^{\mathrm{up}},\Pi_{\mathrm{p}}^{\mathrm{down}}) \leqslant \mathrm{C}\tilde{T}^{\kappa}\} \ .$$

Note that it holds with probability at least $1 - \tilde{T}^{-\zeta} \geqslant 1 - T^{-\alpha\zeta}$ since we suppose that $\Pi_{\mathrm{p}}^{\mathrm{up}}$ satisfies **H2**.

Suppose that we have $\tau_a \geqslant \tau_a^{\star} + 1/T^{\beta}$. By definition of the optimal incentives, we obtain, using by assumption $\tau_a \geqslant \tau_a^{\star} + 1/T^{\beta}$

$$\begin{aligned}
\mathbb{1}_a(a)\tau_a + v^{\mathrm{up}}(a) &\geqslant \tau_a^{\star} + v^{\mathrm{up}}(a) + 1/T^{\beta} \\
&= \max_{a'\in\mathcal{A}} v^{\mathrm{up}}(a') - v^{\mathrm{up}}(a) + v^{\mathrm{up}}(a) + 1/T^{\beta} \\
&= \max_{a'} v^{\mathrm{up}}(a') + 1/T^{\beta} \ ,
\end{aligned}$$

which ensures that $a$ is the optimal arm for the upstream player during the batch $\{k_{a,d}+1,\ldots,k_{a,d}+\tilde{T}\}$ that we consider. In that case, since $a$ is the best arm, by definition of the upstream player's utility, the reward gap $\mathbb{E}[v^{\mathrm{up}}(a) + \tau_a - (v^{\mathrm{up}}(A_t) + \mathbb{1}_a(A_t)\tau_a)]$ for the upstream player at any step $t \in \{k_{a,d}+1,\ldots,k_{a,d}+\tilde{T}\}$ is at least $v^{\mathrm{up}}(a) + \tau_a - \max_{a'\in\mathcal{A}} v^{\mathrm{up}}(a')$, and we obtain

$$\mathrm{C}\tilde{T}^{\kappa} \geqslant \tilde{\mathfrak{R}}_{\mathrm{p}}^{\mathrm{up}}(\{k_{a,d}+1,\ldots,k_{a,d}+\tilde{T}\},\Pi_{\mathrm{p}}^{\mathrm{up}},\Pi_{\mathrm{p}}^{\mathrm{down}}) \geqslant T_{a,d}^{\neq}(\tau_a + v^{\mathrm{up}}(a) - \max_{a'\in\mathcal{A}}\{v^{\mathrm{up}}(a')\}) \ ,$$

by **H2**, $T_{a,d}^{\neq}$ being the number of steps for which a suboptimal arm has been chosen. Therefore, by definition of the optimal incentives, we obtain that

$$\mathrm{C}\tilde{T}^{\kappa} \geqslant T_{a,d}^{\neq}(\tau_a - \tau_a^{\star}) \geqslant T_{a,d}^{\neq}/T^{\beta} \geqslant T_{a,d}^{\neq}\tilde{T}^{-\beta/\alpha} \ ,$$

which gives: $T_{a,d}^{\neq} \leqslant \mathrm{C}\tilde{T}^{\kappa+\beta/\alpha}$. Therefore, if we take the contrapositive, we obtain that during the sequence $\{k_{a,d}+1,\ldots,k_{a,d}+\tilde{T}\}$, if $T_{a,d}^{\neq} > \mathrm{C}\tilde{T}^{\kappa+\beta/\alpha}$, then with probability at least $1 - T^{-\alpha\zeta}$, $\tau_a < \tau_a^{\star} + 1/T^{\beta}$, or equivalently $\tau_a^{\star} > \tau_a - 1/T^{\beta}$.

Now suppose that $\tau_a \leqslant \tau_a^{\star} - 1/T^{\beta}$. By definition of the optimal incentives, we obtain

$$\begin{aligned}
\mathbb{1}_a(a)\tau_a + v^{\mathrm{up}}(a) &\leqslant \tau_a^{\star} + v^{\mathrm{up}}(a) \\
&\leqslant \max_{a'\in\mathcal{A}} v^{\mathrm{up}}(a') - v^{\mathrm{up}}(a) + v^{\mathrm{up}}(a) - 1/T^{\beta} \\
&= \max_{a'} v^{\mathrm{up}}(a') - 1/T^{\beta} \ ,
\end{aligned}$$

which ensures that $a$ is a suboptimal arm for the upstream player during this batch of time steps. Therefore, arm $a$ which has a reward gap bigger than $1/T^{\beta}$ since $\max_{a'\in\mathcal{A}}\{v^{\mathrm{up}}(a) + \mathbb{1}_a(a')\} - (v^{\mathrm{up}}(a) + \tau_a) \geqslant 1/T^{\beta}$ and arm $a$ has been picked $\tilde{T} - T_{a,d}^{\neq}$ times. Consequently, we have that

$$\begin{aligned}
\mathrm{C}\tilde{T}^{\kappa} &\geqslant \tilde{\mathfrak{R}}_{\mathrm{p}}^{\mathrm{up}}(\{k_{a,d}+1,\ldots,k_{a,d}+\tilde{T}\},\Pi_{\mathrm{p}}^{\mathrm{up}},\Pi_{\mathrm{p}}^{\mathrm{down}}) \\
&\geqslant (\tilde{T} - T_{a,d}^{\neq})\underbrace{(\max_{a'\in\mathcal{A}} v^{\mathrm{up}}(a') - (\tau_a + v^{\mathrm{up}}(a)))}_{\geqslant 1/T^{\beta}} \\
&\geqslant (\tilde{T} - T_{a,d}^{\neq})\tilde{T}^{-\beta/\alpha} \ ,
\end{aligned}$$

which gives $T_{a,d}^{\neq} \geqslant \tilde{T} - \mathrm{C}\tilde{T}^{\kappa+/\beta/\alpha}$. Therefore, if we take the contrapositive, we obtain that if $T_{a,d}^{\neq} < \tilde{T} - \mathrm{C}\tilde{T}^{\kappa+\beta/\alpha}$, then with probability at least $1 - T^{-\alpha\zeta}$, $\tau_a > \tau_a^{\star} - 1/T^{\beta}$, or equivalently $\tau_a^{\star} < \tau_a + 1/T^{\beta}$.

For the second part of the proof, suppose that: $\mathrm{C}\tilde{T}^{\kappa+\beta/\alpha} < T_{a,d}^{\neq} < \tilde{T}^{\kappa+\beta/\alpha} - \mathrm{C}\tilde{T}^{\kappa+\beta/\alpha}$.

From the above result, we have that $\tau_a^{\star} < \tau_a + 1/T^{\beta}$ and $\tau_a^{\star} > \tau_a - 1/T^{\beta}$ with probability at least $1 - 2T^{-\alpha\zeta}$. Plugging these inequalities in the absolute value $|\tau_a^{\star} - \tau_a|$ concludes the proof. □

**Lemma 3.** *Assume that we run Algorithm 2 and consider some binary search batch iteration $d \in \lceil \log T^{\beta} \rceil$ run on arm $a \in \mathcal{A}$. Then $0 \leqslant \underline{\tau}_a(d) \leqslant \tau_a^{\mathrm{mid}}(d) \leqslant \overline{\tau}_a(d) \leqslant 1$.*

*Proof of Lemma 3.* The proof proceeds by induction. Considering some action $a \in \mathcal{A}$, we have for the initialisation before any binary search is run: $\underline{\tau}_a(0) = 0, \overline{\tau}_a(0) = 1$ and therefore $\tau_a^{\mathrm{mid}}(0) \in [\underline{\tau}_a(0), \overline{\tau}_a(0)]$. We now consider that a number $d$ of binary search batches has been run on $a$. Suppose that we run an additional binary search batch on action $a$. We have

$$\tau_a^{\mathrm{mid}}(d+1) = \frac{\overline{\tau}_a(d) + \underline{\tau}_a(d)}{2} \text{ which gives } \tau_a^{\mathrm{mid}}(d+1) \in [\underline{\tau}_a(d), \overline{\tau}_a(d)] . \tag{22}$$

After this iteration of binary search, we either have $\overline{\tau}_a(d+1) = \tau_a^{\mathrm{mid}}(d+1) + 1/T^\beta$ and $\underline{\tau}_a(d+1) = \underline{\tau}_a(d)$ or $\overline{\tau}_a(d+1) = \tau_a^{\mathrm{mid}}(d)$ and $\underline{\tau}_a(d+1) = \tau_a^{\mathrm{mid}}(d+1) - 1/T^\beta$. Therefore, we still have $0 \leqslant \underline{\tau}_a(d+1) \leqslant \tau_a^{\mathrm{mid}}(d+1) \leqslant \overline{\tau}_a(d+1) \leqslant 1$, hence the result for any $d \in \lceil \log_2 T^\beta \rceil$ by induction. $\qquad \square$

**Lemma 4.** *Consider some arm $a \in \mathcal{A}$ and suppose that we have run $D \in \mathbb{N}^\star$ batches of binary search of length $\tilde{T}$ on $a$. Then, we have that*

$$\mathbb{P}\left( \bigcap_{d \in [D]} \{\tilde{\mathfrak{R}}_{\mathrm{p}}^{\mathrm{up}}(\{k_{a,d} + 1, \ldots, k_{a,d} + \tilde{T}\}, \Pi_{\mathrm{p}}^{\mathrm{up}}, \Pi_{\mathrm{p}}^{\mathrm{down}}) \leqslant \mathrm{C}\tilde{T}^\kappa\} \right) \geqslant 1 - D/T^{\alpha\zeta} .$$

*Proof of Lemma 4.* First observe that for any batch $d \in \{1, \ldots, D\}$ of binary search run on arm $a$ during steps $\{k_{a,d} + 1, \ldots, k_{a,d} + \tilde{T}\}$, we have by **H**2 that

$$\mathbb{P}\left( \tilde{\mathfrak{R}}_{\mathrm{p}}^{\mathrm{up}}(\{k_{a,d} + 1, \ldots, k_{a,d} + \tilde{T}\}, \Pi_{\mathrm{p}}^{\mathrm{up}}, \Pi_{\mathrm{p}}^{\mathrm{down}}) \geqslant \mathrm{C}\tilde{T}^\kappa \right) \leqslant 1/\tilde{T}^\zeta ,$$

and applying a union bound over the $D$ batches, we have that

$$\mathbb{P}\left( \bigcup_{d \in [D]} \{\tilde{\mathfrak{R}}_{\mathrm{p}}^{\mathrm{up}}(\{k_{a,d} + 1, \ldots, k_{a,d} + \tilde{T}\}, \Pi_{\mathrm{p}}^{\mathrm{up}}, \Pi_{\mathrm{p}}^{\mathrm{down}}) \geqslant \mathrm{C}\tilde{T}^\kappa\} \right)$$

$$\leqslant \sum_{j=1}^{D} \mathbb{P}\left( \{\tilde{\mathfrak{R}}_{\mathrm{p}}^{\mathrm{up}}(\{k_{a,d} + 1, \ldots, k_{a,d} + \tilde{T}\}, \Pi_{\mathrm{p}}^{\mathrm{up}}, \Pi_{\mathrm{p}}^{\mathrm{down}}) \geqslant \mathrm{C}\tilde{T}^\kappa\} \right)$$

$$\leqslant D/\tilde{T}^\zeta ,$$

which gives that

$$\mathbb{P}\left( \bigcap_{d \in [D]} \{\tilde{\mathfrak{R}}_{\mathrm{p}}^{\mathrm{up}}(\{k_{a,d} + 1, \ldots, k_{a,d} + \tilde{T}\}, \Pi_{\mathrm{p}}^{\mathrm{up}}, \Pi_{\mathrm{p}}^{\mathrm{down}}) \leqslant \mathrm{C}\tilde{T}^\kappa\} \right) \geqslant 1 - D/T^{\alpha\zeta} ,$$

hence the result. $\qquad \square$

**Lemma 5.** *Suppose that the upstream player runs a subroutine $\Pi_{\mathrm{p}}^{\mathrm{up}}$ satisfying **H** 2. Consider some action $a \in \mathcal{A}$ and the $D$-th binary search batch of length $\tilde{T}$ run on arm $a$ with $D \in \{1, \ldots, \lceil \log_2 T^\beta \rceil\}$. Then*

$$\bigcap_{d \in [D]} \{\tilde{\mathfrak{R}}_{\mathrm{p}}^{\mathrm{up}}(\{k_{a,d} + 1, \ldots, k_{a,d} + \tilde{T}\}, \Pi_{\mathrm{p}}^{\mathrm{up}}, \Pi_{\mathrm{p}}^{\mathrm{down}}) \leqslant \mathrm{C}\tilde{T}^\kappa\} \subseteq \{\tau_a^\star \in [\underline{\tau}_a(D), \overline{\tau}_a(D)]\} ,$$

*and the probability of these events is at least $1 - \lceil \log_2 T^\beta \rceil / T^{\alpha\zeta}$. We also have that*

$$|\overline{\tau}_a(D) - \underline{\tau}_a(D)| \leqslant 1/2^D + 2/T^\beta \text{ holds almost surely .}$$

*Proof of Lemma 5.* Suppose that the conditions of the lemma hold and consider some arm $a$.

We show by induction on the number of binary search batches $D$ that have been run on $a$ that $\bigcap_{d \in [D]} \{\tilde{\mathfrak{R}}_{\mathrm{p}}^{\mathrm{up}}(\{k_{a,d} + 1, \ldots, k_{a,d} + \tilde{T}\}, \Pi_{\mathrm{p}}^{\mathrm{up}}, \Pi_{\mathrm{p}}^{\mathrm{down}}) \leqslant \mathrm{C}\tilde{T}^\kappa\} \subseteq \{\tau_a^\star \in [\underline{\tau}_a(D), \overline{\tau}_a(D)]\}$. If it is true, Lemma 4 completes this first part of the proof.

The initialisation holds since $\underline{\tau}_a(0) = 0$, $\overline{\tau}_a(0) = 1$ and $\tau_a^\star = \max_{a' \in \mathcal{A}} v^{\mathrm{up}}(a') - v^{\mathrm{up}}(a) \in [0, 1]$ with probability 1 - since $\max_{a' \in \mathcal{A}} v^{\mathrm{up}}(a') \in [0, 1]$ and $v^{\mathrm{up}}(a) \in [0, 1]$.

We suppose that the property is true for some integer $D < \lceil \log_2 T^\beta \rceil$ and that we have run one more binary search on arm $a$. We have

$$\tau_a^{\mathrm{mid}}(D+1) = \frac{\overline{\tau}_a(D) + \underline{\tau}_a(D)}{2} \; ,$$

$\tau_a^{\mathrm{mid}}(D+1)$ being the incentive offered to the upstream player if he chooses action $a$ during the $D$-th batch $\{k_{a,D}+1, \ldots, k_{a,D} + \tilde{T}\}$. After this batch, if $T_{a,D}^{\neq} < \tilde{T} - C\tilde{T}^{\kappa+\beta/\alpha}$, BELGIC updates $\overline{\tau}_a(D+1) = \tau_a^{\mathrm{mid}}(D+1) + 1/T^\beta$, $\underline{\tau}_a(D+1) = \underline{\tau}_a(D)$ and Lemma 2 ensures that $\underline{\tau}_a(D+1) < \tau_a^\star < \overline{\tau}_a(D+1)$ given $\{\tilde{\mathfrak{R}}_{\mathrm{p}}^{\mathrm{up}}(\{k_{a,D+1}+1, \ldots, k_{a,D+1}+\tilde{T}\}, \Pi_{\mathrm{p}}^{\mathrm{up}}, \Pi_{\mathrm{p}}^{\mathrm{down}}) \leqslant C\tilde{T}^\kappa\}$. Thus the induction holds.

Otherwise, if $T_{a,D}^{\neq} > C\tilde{T}^{\kappa+\beta/\alpha}$, BELGIC updates $\underline{\tau}_a(D+1) = \tau_a^{\mathrm{mid}}(D+1) - 1/T^\beta$, $\overline{\tau}_a(D+1) = \overline{\tau}_a(D)$ and Lemma 2 ensures that $\underline{\tau}_a(D+1) < \tau_a^\star < \overline{\tau}_a(D+1)$ given $\{\tilde{\mathfrak{R}}_{\mathrm{p}}^{\mathrm{up}}(\{k_{a,D+1}+1, \ldots, k_{a,D+1}+\tilde{T}\}, \Pi_{\mathrm{p}}^{\mathrm{up}}, \Pi_{\mathrm{p}}^{\mathrm{down}}) \leqslant C\tilde{T}^\kappa\}$. The induction still holds.

Consequently, we have that for any number $D$ of binary search batches run on arm $a$, $\tau_a^\star \in [\underline{\tau}_a(D), \overline{\tau}_a(D)]$ with probability $1 - D/T^{\alpha\zeta}$

For the second part of the proof, we define $u(D) = \overline{\tau}_a(D) - \underline{\tau}_a(D) \geqslant 0$ as the length of the interval containing $\tau_a^\star$ with probability at least $1 - D/T^{\alpha\zeta}$. We have $u(0) = 1$. Suppose that after $D$ iterations of binary search batches, the next batch of binary search $\{k_{a,D+1}+1, \ldots, k_{a,D+1}+\tilde{T}\}$ outputs $T_{a,D+1}^{\neq} < \tilde{T} - C\tilde{T}^{\kappa+\beta/\alpha}$. Then, the update of Algorithm 2 gives

$$\begin{aligned}
u_{D+1} &= \overline{\tau}_a(D+1) - \underline{\tau}_a(D+1) \\
&= \tau_a^{\mathrm{mid}}(D+1) + 1/T^\beta - \underline{\tau}_a(D) \\
&= \frac{\overline{\tau}_a(D) + \underline{\tau}_a(D)}{2} - \underline{\tau}_a(D) + 1/T^\beta \\
&= \frac{\overline{\tau}_a(D) - \underline{\tau}_a(D)}{2} + 1/T^\beta \\
&= u_D/2 + 1/T^\beta \; .
\end{aligned}$$

On the other hand, if $T_{a,D+1}^{\neq} > C\tilde{T}^{\kappa+\beta/\alpha}$, the update gives

$$\begin{aligned}
u_{D+1} &= \overline{\tau}_a(D+1) - \underline{\tau}_a(D+1) \\
&= \overline{\tau}_a(D_t^a) - (\tau_a^{\mathrm{mid}}(D+1) - 1/T^\beta) \\
&= \overline{\tau}_a(D) - \frac{\overline{\tau}_a(D) + \underline{\tau}_a(D)}{2} + 1/T^\beta \\
&= \frac{\overline{\tau}_a(D) - \underline{\tau}_a(D)}{2} + 1/T^\beta \\
&= u_D/2 + 1/T^\beta \; .
\end{aligned}$$

We can see that $(u_D)_{D \geqslant 0}$ is an arithmetico-geometric sequence defined by $u_{D+1} = u_D/2 + 1/T^\beta$ with an initial term $u_0 = 1$. Writing $r = 1/T^\beta/(1 - 1/2) = 2/T^\beta$, we obtain that

$$|\overline{\tau}_a(D) - \underline{\tau}_a(D)| = u_D = 1/2^D(1 - r) + r = 1/2^D(1 - 2/T^\beta) + 2/T^\beta \leqslant 1/2^D + 2/T^\beta \; ,$$

for any $D \in \{1, \ldots, \lceil \log_2 T^\beta \rceil\}$, hence the result. $\qquad\square$

**Lemma 6.** *Suppose that the upstream player runs a policy $\Pi_{\mathrm{p}}^{\mathrm{up}}$ satisfying **H2**. Considering some action $a \in \mathcal{A}$, we have that after the binary search batch $D = \lceil \log_2 T^\beta \rceil$: $\mathbb{P}(\underline{\tau}_a(D) \leqslant \tau_a^\star \leqslant \overline{\tau}_a(D) \leqslant \underline{\tau}_a + 3/T^\beta) \geqslant 1 - D/T^{\alpha\zeta}$.*

*Proof of Lemma 6.* We suppose that the event $\bigcap_{d \in [[\log_2 T^\beta]]}\{\tilde{\mathfrak{R}}_{\mathrm{p}}^{\mathrm{up}}(\{k_{a,d}+1, \ldots, k_{a,d}+\tilde{T}\}, \Pi_{\mathrm{p}}^{\mathrm{up}}, \Pi_{\mathrm{p}}^{\mathrm{down}}) \leqslant C\tilde{T}^\kappa\}$ holds. Lemma 4 ensures that this event holds with probability at least $1 - \lceil \log_2 T^\beta \rceil/T^{\alpha\zeta}$.

After $D = \lceil \beta \log_2 T \rceil$ batches of binary search on arm $a$, we have by Lemma 5 that

$$|\overline{\tau}_a(D) - \underline{\tau}_a(D)| \leqslant 1/2^D + 2/T^\beta \leqslant 1/2^{\beta \log_2 T} + 2/T^\beta = 3/T^\beta \ .$$

Lemma 5 guarantees that $\tau_a^\star \in [\underline{\tau}_a(D); \overline{\tau}_a(D)]$ with probability at least $1 - D/T^{\alpha\zeta}$, and we obtain $\underline{\tau}_a(D) \leqslant \tau_a^\star \leqslant \overline{\tau}_a(D) \leqslant \underline{\tau}_a(D) + 3/T^\beta$ with the same probability. $\qquad\square$

**Proposition 1.** *Under **H1** and **H2**, after the first phase of* `BELGIC` *which consists in* $K\lceil T^\alpha \rceil \lceil \log T^\beta \rceil$ *steps of binary search grouped in* $\lceil \log_2 T^\beta \rceil$ *batches per arm* $a \in \mathcal{A}$*, we have that*

$$\mathbb{P}\Big( \text{ for any } a \in \mathcal{A}, \hat{\tau}_a - 4/T^\beta - CT^{(\kappa-1)/2} \leqslant \tau_a^\star \leqslant \hat{\tau}_a \Big) \geqslant 1 - K\lceil \log_2 T^\beta \rceil/T^{\alpha\zeta} \ .$$

*Proof of Proposition 1.* We consider a number of binary search batches $D = \lceil \log T^\beta \rceil$ and we define the event $\mathcal{G}$ as $\mathcal{G} = \{ \text{for any } a \in \mathcal{A}, \underline{\tau}_a(D) \leqslant \tau_a^\star \leqslant \overline{\tau}_a(D) \leqslant \underline{\tau}_a(D) + 3/T^\beta \}$. We have that

$$\mathbb{P}(\mathcal{G}) = \mathbb{P}\left( \bigcap_{a \in \mathcal{A}} \big\{ \underline{\tau}_a(D) \leqslant \tau_a^\star \leqslant \overline{\tau}_a(D) \leqslant \underline{\tau}_a + 3/T^\beta \big\} \right)$$

$$= 1 - \mathbb{P}\left( \bigcup_{a \in \mathcal{A}} \big\{ \underline{\tau}_a(D) \leqslant \tau_a^\star \leqslant \overline{\tau}_a(t) \leqslant \underline{\tau}_a(D) + 3/T^\beta \big\}^{\mathrm{c}} \right)$$

$$\geqslant 1 - \sum_{a \in \mathcal{A}} \mathbb{P}\Big( \big\{ \underline{\tau}_a(D) \leqslant \tau_a^\star \leqslant \overline{\tau}_a(D) \leqslant \underline{\tau}_a(D) + 3/T^\beta \big\}^{\mathrm{c}} \Big) \ ,$$

where the last inequality holds with an union bound. Lemma 6 with $D = \lceil \log_2 T^\beta \rceil$ ensures that we have

$$\mathbb{P}\Big( \big\{ \underline{\tau}_a(D) \leqslant \tau_a^\star \leqslant \overline{\tau}_a(D) \leqslant \underline{\tau}_a(D) + 3/T^\beta \big\}^{\mathrm{c}} \Big) \leqslant D/T^{\alpha\zeta} \ ,$$

and since $\mathrm{Card}\{\mathcal{A}\} = K$, we obtain

$$\mathbb{P}(\mathcal{G}) \geqslant 1 - K\lceil \log_2 T^\beta \rceil/T^{\alpha\zeta} \ .$$

Since the estimated incentives are defined as $\hat{\tau}_a = \overline{\tau}_a(\lceil \log_2 T^\beta \rceil) + 1/T^\beta + CT^{(\kappa-1)/2}$, we can conclude

$$\mathbb{P}( \text{ for any } a \in \mathcal{A}, \hat{\tau}_a - 4/T^\beta - CT^{(\kappa-1)/2} \leqslant \tau_a^\star \leqslant \hat{\tau}_a) \geqslant 1 - K\lceil \log_2 T^\beta \rceil/T^{\alpha\zeta} \ ,$$

since whenever $\underline{\tau}_a(\lceil \log_2 T^\beta \rceil) \leqslant \tau_a^\star \leqslant \overline{\tau}_a(\lceil \log_2 T^\beta \rceil) \leqslant \underline{\tau}_a(\lceil \log_2 T^\beta \rceil) + 1/T^\beta$, we also have by definition: $\hat{\tau}_a(\lceil \log_2 T^\beta \rceil) - 4/T^\beta - CT^{(\kappa-1)/2} \leqslant \underline{\tau}_a(\lceil \log_2 T^\beta \rceil) \leqslant \tau_a^\star \leqslant \hat{\tau}_a(\lceil \log_2 T^\beta \rceil)$. $\qquad\square$

**Theorem 3.** *Assume that **H1** and **H2** hold. Then* `BELGIC`*, run with* $\alpha, \beta$ *satisfying* (13) *and any bandit subroutine* `Bandit-Alg`*, has an overall regret* $\mathfrak{R}_{\mathrm{p}}^{\mathrm{down}}$ *such that*

$$\mathfrak{R}_{\mathrm{p}}^{\mathrm{down}}(T, \Pi_{\mathrm{p}}^{\mathrm{up}}, \texttt{BELGIC}) \leqslant 2(3 + 2C + \bar{v} - \underline{v})\log_2(T)(2T^{1-\alpha\zeta} + T^{(\kappa+1)/2} + \lceil T^\alpha \rceil) + 4T^{1-\beta}$$
$$+ \mathfrak{R}_{\texttt{Bandit-Alg}}(T, \nu, \{\hat{\tau}_a\}_{a \in \mathcal{A}})$$

*where, for ease of notation*

$$\bar{v} = \max_{a,b \in \mathcal{A} \times \mathcal{A}} \{v^{\mathrm{down}}(a,b)\} \quad \text{and} \quad \underline{v} = \min_{a,b \in \mathcal{A} \times \mathcal{A}} \{v^{\mathrm{down}}(a,b)\} \ .$$

*Proof of Theorem 3.* Suppose that the conditions of Theorem 3 are satisfied. By definition, $\Lambda_{K+1} + 1 \in [T]$ is the step at which starts the run of the subroutine `Bandit-Alg`, since $\Lambda_{K+1} = K\lceil T^\alpha \rceil \lceil \beta \log T \rceil$. All the binary searche batches have length $\tilde{T} = \lceil T^\alpha \rceil$. For any $a \in \mathcal{A}, d \in \{1, \ldots, \lceil \log_2 T^\beta \rceil\}$, we define the event

$$\mathsf{B}_{a,d} = \Big\{ \tilde{\mathfrak{R}}_{\mathrm{p}}^{\mathrm{up}}(\{k_{a,d}+1, \ldots, k_{a,d} + \tilde{T}\}, \Pi_{\mathrm{p}}^{\mathrm{up}}, \Pi_{\mathrm{p}}^{\mathrm{down}}) \leqslant C\tilde{T}^\kappa \Big\} \ ,$$

as well as

$$\mathcal{E} = \bigcap_{\substack{a \in [K] \\ d \in [\lceil \beta \log T \rceil]}} \mathsf{B}_{a,d} \bigcap \Big\{ \tilde{\mathfrak{R}}_{\mathrm{p}}^{\mathrm{up}}(\{\Lambda_{K+1}+1, \ldots, T\}, \Pi_{\mathrm{p}}^{\mathrm{up}}) \leqslant C(T - \Lambda_{K+1})^\kappa \Big\} \ ,$$

and by **H2** and Lemma 4, with an union bound, we have that

$$\mathbb{P}(\mathcal{E}) = 1 - \mathbb{P}\left(\bigcup_{\substack{a \in [K] \\ d = \in [\lceil \log T^\beta \rceil]}} \mathsf{B}_{a,d}^{\mathrm{c}} \bigcup \left\{ \tilde{\mathfrak{R}}_{\mathrm{p}}^{\mathrm{up}}(\{\Lambda_{K+1}+1,\dots,T\}, \Pi_{\mathrm{p}}^{\mathrm{up}}, \Pi_{\mathrm{p}}^{\mathrm{down}}) \leqslant \mathrm{C}(T - \Lambda_{K+1})^\kappa \right\}\right)$$

$$\geqslant 1 - \sum_{\substack{a \in [K] \\ d \in [\lceil \log T^\beta \rceil]}} \mathbb{P}(\mathsf{B}_{a,d}^{\mathrm{c}}) + \mathbb{P}(\tilde{\mathfrak{R}}_{\mathrm{p}}^{\mathrm{up}}(\{\Lambda_{K+1}+1,\dots,T\}, \Pi_{\mathrm{p}}^{\mathrm{up}}) \leqslant \mathrm{C}(T - \Lambda_{K+1})^\kappa)$$

$$\geqslant 1 - K\lceil \beta \log T \rceil \tilde{T}^{-\zeta} - (T - \Lambda_{K+1})^{-\zeta}$$

$$\geqslant 1 - K\lceil \beta \log T \rceil T^{-\alpha\zeta} - T^{-\zeta} \ ,$$

and we now decompose

$$\mathfrak{R}_{\mathrm{p}}^{\mathrm{down}}(T, \Pi_{\mathrm{p}}^{\mathrm{up}}, \Pi_{\mathrm{p}}^{\mathrm{down}}) = \mathbb{E}\left[\mathbb{1}(\mathcal{E})\tilde{\mathfrak{R}}_{\mathrm{p}}^{\mathrm{down}}(T, \Pi_{\mathrm{p}}^{\mathrm{up}}, \Pi_{\mathrm{p}}^{\mathrm{down}}) + \left[\mathbb{1}(\mathcal{E}^{\mathrm{c}})\tilde{\mathfrak{R}}_{\mathrm{p}}^{\mathrm{down}}(T, \Pi_{\mathrm{p}}^{\mathrm{up}}, \Pi_{\mathrm{p}}^{\mathrm{down}})\right] \ , \tag{23}$$

where $\tilde{\mathfrak{R}}_{\mathrm{p}}^{\mathrm{down}}$ is defined as

$$\tilde{\mathfrak{R}}_{\mathrm{p}}^{\mathrm{down}}(T, \Pi_{\mathrm{p}}^{\mathrm{up}}, \Pi_{\mathrm{p}}^{\mathrm{down}}) = T\mu^{\star,\mathrm{down}} - \sum_{t=1}^{T}(v^{\mathrm{down}}(A_t, B_t) - \mathbb{1}_{\tilde{a}_t}(A_t)\tau(t)) \ .$$

By definition, $\mu^{\star,\mathrm{down}} \geqslant \min_{a,b \in \mathcal{A} \times \mathcal{A}} v^{\mathrm{down}}(a, b) \geqslant \underline{v}$ and since Lemma 1 allows to write $\mu^{\star,\mathrm{down}} = \max_{a,b \in \mathcal{A} \times \mathcal{A}}\{v^{\mathrm{down}}(a, b) + v^{\mathrm{up}}(a)\} - \max_{a' \in \mathcal{A}}\{v^{\mathrm{up}}(a')\}$, we have that

$$\underline{v} \leqslant \mu^{\star,\mathrm{down}} \leqslant 1 + \bar{v} \ ,$$

and note that since $\kappa < 1$, for any $a \in \mathcal{A}, \hat{\tau}_a \leqslant 2 + \mathrm{C}T^{(\kappa-1)/2} \leqslant 2 + \mathrm{C}$. Consequently, $T\underline{v} \leqslant \tilde{\mathfrak{R}}_{\mathrm{p}}^{\mathrm{down}}(T, \Pi_{\mathrm{p}}^{\mathrm{up}}, \Pi_{\mathrm{p}}^{\mathrm{down}}) \leqslant (3 + \mathrm{C} + \bar{v} - \underline{v})T$ almost surely. Therefore

$$\mathbb{E}\left[\mathbb{1}(\mathcal{E}^{\mathrm{c}})\tilde{\mathfrak{R}}_{\mathrm{p}}^{\mathrm{down}}(T, \Pi_{\mathrm{p}}^{\mathrm{up}}, \Pi_{\mathrm{p}}^{\mathrm{down}})\right] \leqslant (3 + \mathrm{C} + \bar{v} - \underline{v})(K\lceil \log T^\beta \rceil T^{1-\alpha\zeta} + T^{1-\zeta}) \ . \tag{24}$$

We consider the second term in (23). We decompose it between the steps of binary search $\{1, \dots, K\lceil T^\alpha \rceil \lceil \log_2 T^\beta \rceil\}$ during which we run the `Binary Search Subroutine` and the following ones when we run `Bandit-Alg`, which gives

$$\mathbb{E}\left[\mathbb{1}(\mathcal{E})\tilde{\mathfrak{R}}_{\mathrm{p}}^{\mathrm{down}}(T, \Pi_{\mathrm{p}}^{\mathrm{up}}, \Pi_{\mathrm{p}}^{\mathrm{down}})\right] \tag{25}$$

$$\leqslant \mathbb{E}\left[\mathbb{1}(\mathcal{E})\underbrace{\sum_{t=1}^{K\lceil T^\alpha \rceil \lceil \log_2 T^\beta \rceil} \mu^{\star,\mathrm{down}} - (v^{\mathrm{down}}(A_t, B_t) - \mathbb{1}_{\tilde{a}_t}(A_t)\tau(t))}_{(\mathbf{A})}\right]$$

$$+ \mathbb{E}\left[\mathbb{1}(\mathcal{E})\underbrace{\sum_{t=K\lceil T^\alpha \rceil \lceil \log_2 T^\beta \rceil+1}^{T} \mu^{\star,\mathrm{down}} - (v^{\mathrm{down}}(A_t, B_t) - \mathbb{1}_{\tilde{a}_t}(A_t)\tau(t))}_{(\mathbf{B})}\right] \ .$$

Similarly to (24), we use the bound on $\mu^{\star,\mathrm{down}}$ to bound $(\mathbf{A})$, which gives

$$\mathbb{E}[\mathbb{1}(\mathcal{E})(\mathbf{A})] \leqslant \mathbb{E}\left[\mathbb{1}(\mathcal{E})\sum_{t=1}^{\Lambda_{K+1}} 1 + \bar{v} - (\underline{v} - 2 - \mathrm{C})\right] \leqslant K\lceil T^\alpha \rceil (1 + \log_2 T)(3 + \mathrm{C} + \bar{v} - \underline{v}) \ . \tag{26}$$

After the binary search, at each step $t \in \{\Lambda_{K+1} + 1, \ldots, T\}$, `Bandit-Alg` recommends $(\tilde{a}_t, B_t) \in \mathcal{A} \times \mathcal{A}$ following (15) and `BELGIC` offers an incentive $(\tilde{a}_t, \hat{\tau}_{\tilde{a}_t})$ with $\hat{\tau}_a = \overline{\tau}_a(\lceil \log_2 T^\beta \rceil) + 1/T^\beta + CT^{(\kappa-1)/2}$.

Lemma 5 ensures that $\mathcal{E} \subseteq \{\tau_a^\star \in [\underline{\tau}_a(\lceil \log_2 T^\beta \rceil), \overline{\tau}_a(\lceil \log_2 T^\beta \rceil)]$ for any $a \in \mathcal{A}\} \bigcap \{\tilde{\mathfrak{R}}_{\mathrm{p}}^{\mathrm{up}}(\{\Lambda_{K+1} + 1, \ldots, T\}, \Pi_{\mathrm{p}}^{\mathrm{up}}) \leqslant C(T - \Lambda_{K+1})^\kappa\}$. Therefore, if $\mathcal{E}$ holds, for any $a \in \mathcal{A}$, we have that

$$
\begin{aligned}
v^{\mathrm{up}}(a) + \hat{\tau}_a &= v^{\mathrm{up}}(a) + \overline{\tau}_a + 1/T^\beta + CT^{(\kappa-1)/2} \\
&> v^{\mathrm{up}}(a) + \tau_a^\star + CT^{(\kappa-1)/2} \\
&= v^{\mathrm{up}}(a) + \max_{a'' \in \mathcal{A}} v^{\mathrm{up}}(a'') - v^{\mathrm{up}}(a) + CT^{(\kappa-1)/2} \,,
\end{aligned}
$$

and therefore, for any $a' \in \mathcal{A}$ such that $a' \neq a$, we have that

$$
v^{\mathrm{up}}(a) + \hat{\tau}_a > v^{\mathrm{up}}(a') + CT^{(\kappa-1)/2} \,. \tag{27}
$$

This shows that at any steps $t \in \{\Lambda_{K+1} + 1, \ldots, T\}$ after the binary search, we have on the event $\mathcal{E}$ that

$$
\tilde{a}_t = \mathrm{argmax}_{a' \in \mathcal{A}} \{v^{\mathrm{up}}(a') + \mathbb{1}_{\tilde{a}_t}(a')\hat{\tau}_{\tilde{a}_t}\} \,, \tag{28}
$$

and the reward gap at step $t$ for any $a \neq \tilde{a}_t$ is defined as

$$
\max_{a' \in \mathcal{A}} \{v^{\mathrm{up}}(a') + \mathbb{1}_{\tilde{a}_t}(a')\hat{\tau}_{\tilde{a}_t}\} - (v^{\mathrm{up}}(a) + \mathbb{1}_{\tilde{a}_t}(a)\hat{\tau}_{\tilde{a}_t}) = v^{\mathrm{up}}(\tilde{a}_t) + \hat{\tau}_a - v^{\mathrm{up}}(a) \,. \tag{29}
$$

Following (27), the reward gap from (29) satisfies

$$
\max_{a' \in \mathcal{A}} \{v^{\mathrm{up}}(a') + \mathbb{1}_{\tilde{a}_t}(a')\hat{\tau}_{\tilde{a}_t}\} - (v^{\mathrm{up}}(a) + \mathbb{1}_{\tilde{a}_t}(a)\hat{\tau}_{\tilde{a}_t}) \geqslant CT^{(\kappa-1)/2} \,.
$$

We now define two sets

$$
\begin{aligned}
\mathrm{I}_T &= \{t \in \{K\lceil T^\alpha \rceil \lceil \log_2 T \rceil + 1, \ldots, T\} \text{ such that } \tilde{a}_t = A_t\} \,, \\
\mathrm{J}_T &= \{t \in \{K\lceil T^\alpha \rceil \lceil \log_2 T \rceil + 1, \ldots, T\} \text{ such that } \tilde{a}_t \neq A_t\} \,,
\end{aligned}
$$

which satisfy $\mathrm{I}_T \cup \mathrm{J}_T = \{K\lceil T^\alpha \rceil \lceil \log_2 T \rceil + 1, \ldots, T\}$ almost surely. As shown in (28), $\mathrm{I}_T$ corresponds to all the steps during which the upstream player picked the best arm and for any $t \in \mathrm{I}_T$

$$
v^{\mathrm{up}}(A_t) + \mathbb{1}_{\tilde{a}_t}(A_t)\tau(t) \geqslant \max_{a \in \mathcal{A}} \{v^{\mathrm{up}}(a) + \mathbb{1}_{\tilde{a}_t}(a)\tau(t)\} \,,
$$

while by (29), for any $t \in \mathrm{J}_T$, we have that

$$
\max_{a \in \mathcal{A}} \{v^{\mathrm{up}}(a) + \mathbb{1}_{\tilde{a}_l}(a)\hat{\tau}_{\tilde{a}_l}\} - (v^{\mathrm{up}}(A_l) + \mathbb{1}_{\tilde{a}_l}(A_l)\hat{\tau}_{\tilde{a}_l}) \geqslant CT^{(\kappa-1)/2} \,. \tag{30}
$$

**H2** ensures that if $\mathcal{E}$ holds, then $\mathfrak{R}_{\mathrm{p}}^{\mathrm{up}}(\{\Lambda_{K+1} + 1, \ldots, T\}, \Pi_{\mathrm{p}}^{\mathrm{up}}, \text{BELGIC}) \leqslant C\,T^\kappa$, and this condition together with (30) gives that

$$
\mathrm{Card}\{\mathrm{J}_T\}\, C\, T^{(\kappa-1)/2} \leqslant \mathfrak{R}_{\mathrm{p}}^{\mathrm{up}}(\{\Lambda_{K+1} + 1, \ldots, T\}, \Pi_{\mathrm{p}}^{\mathrm{up}}, \text{BELGIC}) \leqslant CT^\kappa \,,
$$

and consequently $\text{Card}\{\text{J}_T\} \leqslant T^{(\kappa+1)/2}$. We now bound $(\mathbf{B})$ as follows

$$
\mathbb{E}[\mathbb{1}(\mathcal{E})(\mathbf{B})] = \mathbb{E}\left[\mathbb{1}(\mathcal{E}) \sum_{t=\Lambda_{K+1}+1}^{T} \mu^{\star,\text{down}} - \left(v^{\text{down}}(A_t, B_t) - \hat{\tau}_{\tilde{a}_t}\right)\right]
$$

$$
= \mathbb{E}\left[\mathbb{1}(\mathcal{E}) \sum_{t\in\text{I}_T} \max_{a,b\in\mathcal{A}\times\mathcal{A}}\{v^{\text{down}}(a,b) - \tau_a^{\star}\} - \left(v^{\text{down}}(\tilde{a}_t, B_t) - \hat{\tau}_{\tilde{a}_t}\right)\right]
$$

$$
+ \mathbb{E}\left[\mathbb{1}(\mathcal{E}) \sum_{t\in\text{J}_T} \underbrace{\mu^{\star,\text{down}} - \left(v^{\text{down}}(A_t, B_t) - \hat{\tau}_{\tilde{a}_t}\right)}_{\leqslant 3+\text{C}+\bar{v}-\underline{v}}\right]
$$

$$
\leqslant \mathbb{E}\left[\mathbb{1}(\mathcal{E}) \sum_{t\in\text{I}_T} \max_{a,b\in\mathcal{A}\times\mathcal{A}}\{v^{\text{down}}(a,b) - \hat{\tau}_a - (v^{\text{down}}(\tilde{a}_t, B_t) - \hat{\tau}_{\tilde{a}_t})\} + \max_{a'\in\mathcal{A}}\{\hat{\tau}_{a'} - \tau_{a'}^{\star}\}\right]
$$

$$
+ (3 + \text{C} + \bar{v} - \underline{v})\mathbb{E}[\mathbb{1}(\mathcal{E})\text{Card}\{\text{J}_T\}]
$$

$$
= \mathbb{E}\left[\mathbb{1}(\mathcal{E}) \sum_{t\in\text{I}_T} \max_{a,b\in\mathcal{A}\times\mathcal{A}}\{v^{\text{down}}(a,b) - \hat{\tau}_a\} - (v^{\text{down}}(\tilde{a}_t, B_t) - \hat{\tau}_{\tilde{a}_t})\right]
$$

$$
+ \mathbb{E}\left[\mathbb{1}(\mathcal{E}) \sum_{t\in\text{I}_T} \max_{a'\in\mathcal{A}}\{\hat{\tau}_{a'} - \tau_{a'}^{\star}\}\right] + (3 + \text{C} + \bar{v} - \underline{v})\,T^{(\kappa+1)/2}
$$

$$
\leqslant \mathfrak{R}_{\texttt{Bandit-Alg}}(\text{Card}\{\text{I}_T\}, \nu, \{\hat{\tau}_a\}_{a\in\mathcal{A}}) + \mathbb{E}\left[\mathbb{1}(\mathcal{E})\text{Card}\{\text{I}_T\} \max_{a'\in\mathcal{A}}\{\hat{\tau}_{a'} - \tau_{a'}^{\star}\}\right]
$$

$$
+ (3 + \text{C} + \bar{v} - \underline{v})\,T^{(\kappa+1)/2}\,,
$$

where the first step holds by Lemma 1. Using Lemma 5, as well as the definition of $\mathcal{E}$, we obtain

$$
\mathbb{E}\left[\mathbb{1}(\mathcal{E})\text{Card}\{\text{I}_T\} \max_{a'\in\mathcal{A}}\{\hat{\tau}_{a'} - \tau_{a'}^{\star}\}\right] \leqslant \mathbb{E}\left[\mathbb{1}(\mathcal{E})\text{Card}\{\mathcal{E}\}(4/T^{\beta} + \text{C}T^{(\kappa-1)/2})\right]
$$

$$
\leqslant \mathbb{E}[\mathbb{1}(\mathcal{E})T](4/T^{\beta} + \text{C}T^{(\kappa-1)/2})
$$

$$
\leqslant 4T^{1-\beta} + \text{C}T^{(\kappa+1)/2}\,,
$$

which finally gives

$$
\mathbb{E}[\mathbb{1}(\mathcal{E})(\mathbf{B})] \leqslant \mathfrak{R}_{\texttt{Bandit-Alg}}(T, \nu, \{\hat{\tau}_a\}_{a\in\mathcal{A}}) + 4T^{1-\beta} + (3 + 2\text{C} + \bar{v} - \underline{v})\,T^{(1+\kappa)/2}\,, \qquad (31)
$$

and plugging together (26) and (31) in the decomposition (25) gives the following bound

$$
\mathbb{E}\left[\mathbb{1}(\mathcal{E})\tilde{\mathfrak{R}}_{\text{p}}^{\text{down}}(T, \Pi_{\text{p}}^{\text{up}}, \texttt{BELGIC}))\right] \leqslant \mathfrak{R}_{\texttt{Bandit-Alg}}(T, \nu, \{\hat{\tau}_a\}_{a\in\mathcal{A}}) + 4T^{1-\beta} + (3 + 2\text{C} + \bar{v} - \underline{v})\,T^{(1+\kappa)/2}
$$

$$
+ (3 + \text{C} + \bar{v} - \underline{v})K\lceil T^{\alpha}\rceil(1 + \log_2 T)\,,
$$

and summing the bounds on the events $\mathcal{E}$ and $\mathcal{E}^{\text{c}}$ finally gives

$$
\mathfrak{R}_{\text{p}}^{\text{down}}(T, \Pi_{\text{p}}^{\text{up}}, \texttt{BELGIC}) \leqslant \mathfrak{R}_{\texttt{Bandit-Alg}}(T, \nu, \{\hat{\tau}_a\}_{a\in\mathcal{A}}) + 4T^{1-\beta} + (3 + 2\text{C} + \bar{v} - \underline{v})\,T^{(1+\kappa)/2}
$$

$$
+ (2 + \text{C} + \bar{v} - \underline{v})K\lceil T^{\alpha}\rceil(1 + \log_2 T)
$$

$$
+ (3 + \text{C} + \bar{v} - \underline{v})(K(1 + \log_2 T)T^{1-\alpha\zeta} + T^{1-\zeta})
$$

$$
\leqslant \mathfrak{R}_{\texttt{Bandit-Alg}}(T, \nu, \{\hat{\tau}_a\}_{a\in\mathcal{A}}) + 4T^{1-\beta} + (3 + 2\text{C} + \bar{v} - \underline{v})\,T^{(1+\kappa)/2}
$$

$$
+ (3 + \bar{C} + \bar{v} - \underline{v})((1 + \log_2 T)(\lceil T^{\alpha}\rceil + T^{1-\alpha\zeta}) + T^{1-\zeta})
$$

$$
\leqslant (3 + 2\text{C} + \bar{v} - \underline{v})(T^{1-\zeta} + T^{(\kappa+1)/2} + (1 + \log_2 T)(\lceil T^{\alpha}\rceil + T^{1-\alpha\zeta}))
$$

$$
+ \mathfrak{R}_{\texttt{Bandit-Alg}}(T, \nu, \{\hat{\tau}_a\}_{a\in\mathcal{A}}) + 4T^{1-\beta}
$$

$$
\leqslant 2(3 + 2\text{C} + \bar{v} - \underline{v})\log_2(T)(2T^{1-\alpha\zeta} + T^{(\kappa+1)/2} + \lceil T^{\alpha}\rceil) + 4T^{1-\beta}
$$

$$
+ \mathfrak{R}_{\texttt{Bandit-Alg}}(T, \nu, \{\hat{\tau}_a\}_{a\in\mathcal{A}})\,.
$$

$\square$

**Corollary 1.** *Assume that the upstream player's distribution $(\gamma_a)_{a\in\mathcal{A}}$ is such that **H** 1 holds. In addition, suppose that the distributions $(\gamma_a)_{a\in\mathcal{A}}$ and $(\nu_{a,b})_{a,b\in\mathcal{A}\times\mathcal{A}}$ are 1-sub-Gaussian and that the upstream player plays $\Pi_\mathrm{p}^\mathrm{up} = $ Algorithm 3 (a slight modification of UCB to take into account the incentives). Then the downstream player's regret when she runs* BELGIC *with parameters $\alpha = 3/4$ and $\beta = 1/4$ (which satisfy (13)) and subroutine* Bandit-Alg $=$ UCB *satisfies the following upper bound*[4]

$$\mathfrak{R}_\mathrm{p}^\mathrm{down}(T, \textit{UCB}, \texttt{BELGIC}) \leqslant (10 + 4K + 32\sqrt{K\log_2(KT^3)} + \bar{v} - \underline{v})\log_2(T)(3 + 2T^{3/4})$$
$$+ 3K^2(\bar{v} - \underline{v}) .$$

*Proof of Corollary 1.* First note that $\Pi_\mathrm{p}^\mathrm{up} = $ Algorithm 3 satisfies **H2** with constants $\kappa = 1/2$, $\zeta = 2$, $\mathrm{C} = 8\sqrt{K\log(KT^3)}$, following Proposition 2. Note that $\beta/\alpha = 1/3 < 1/2 = 1 - \kappa$, therefore Equation (13) is satisfied. Plugging these terms in the bound from Theorem 3 with $\alpha = 3/4, \beta = 1/4$ gives

$$\mathfrak{R}_\mathrm{p}^\mathrm{down}(T, \texttt{UCB}, \texttt{BELGIC}) \leqslant 2(3 + 16\sqrt{K\log_2(KT^3)} + \bar{v} - \underline{v})\log_2(T)(2T^{1-3/2} + T^{3/4} + \lceil T^{3/4}\rceil)$$
$$+ 4T^{1/4} + 8\sqrt{K^2\log_2(T)}\,T^{1/2} + 3K^2(\bar{v} - \underline{v})$$

where we use the bound for $\mathfrak{R}_{\texttt{Bandit-Alg}}(T, \nu, \{\hat{\tau}_a\}_{a\in\mathcal{A}})$ with Bandit-Alg $=$ UCB run on any bandit instance with $K^2$ arms, 1-subgaussian rewards and reward gaps of at most $\max_{a,b\in\mathcal{A}\times\mathcal{A}} v^\mathrm{down}(a,b) - \min_{a,b\in\mathcal{A}\times\mathcal{A}} v^\mathrm{down}(a,b) = \bar{v} - \underline{v}$, following [Lattimore and Szepesvári, 2020, Theorem 7.2]. Therefore, we have that

$$\mathfrak{R}_\mathrm{p}^\mathrm{down}(T, \texttt{UCB}, \texttt{BELGIC}) \leqslant (6 + 32\sqrt{K\log_2(KT^3)} + \bar{v} - \underline{v})\log_2(T)(2 + 1 + 2T^{3/4})$$
$$+ 4T^{1/4} + 3K^2(\bar{v} - \underline{v}) + 8K\sqrt{\log_2 T}\,T^{1/2}$$
$$\leqslant (10 + 32\sqrt{K\log_2(KT^3)} + \bar{v} - \underline{v})\log_2(T)(3 + 2T^{3/4})$$
$$+ 3K^2(\bar{v} - \underline{v}) + 8K\sqrt{\log_2(T)}\,T^{1/2} ,$$

which finally gives

$$\mathfrak{R}_\mathrm{p}^\mathrm{down}(T, \texttt{UCB}, \texttt{BELGIC}) \leqslant (10 + 4K + 32\sqrt{K\log_2(KT^3)} + \bar{v} - \underline{v})\log_2(T)(3 + 2T^{3/4})$$
$$+ 3K^2(\bar{v} - \underline{v}) ,$$

hence the result. $\qquad\qquad\square$

---

[4]Note that it is $K$ and not $\sqrt{K}$ here, since the action space is of cardinality $K^2$ for the downstream player.

