# OpenReview forum: "Learning to Mitigate Externalities: the Coase Theorem with Hindsight Rationality"
_NeurIPS.cc/2024/Conference — NeurIPS 2024 spotlight_

### Official Review · Reviewer_UVPQ · 2024-07-12

**Soundness:** 4
**Presentation:** 4
**Contribution:** 3
**Rating:** 7
**Confidence:** 2

**Summary:**

This work considers a two player game with externalities imposed from one agent (“upstream”) to the other (“downstream”) agent. Equilibrium in the absence of taxation or payments between players would result in heavy efficiency loss due to the externalities.

The Coase theorem tells us that in these situations we should expect to see payments from the downstream agent to the upstream agent in order to restore the socially optimal outcome.

This work formulates this scenario as a multi-armed bandit problem, in which the payment term forms a part of the players payoffs and options. Namely, the downstream agent can put a conditional payment on the arms played by the upstream bandit. In this situation, the paper asks whether or not normal bandit learning algorithms can learn the optimal outcome.

The paper puts forwards an assumption that must be satisfied on the upstream agents learning behavior, which is satisfied for many common bandit learning algorithms. A translation is given for the downstream agent to go from the raw bandit problem to a transformed bandit problem. With these, sublinear regret is realized relative to the socially optimal outcome.

**Strengths:**

The work takes a well-known economic theory and asks whether a given type of ML can recover the expected phenomenon without knowing the specific externality a prior. Rather than showing an exact algorithm for recovering it, mappings are done so that any suitable bandit learning algorithm can successfully learn the results. This is a nice addition over prior work, and helps in suggesting that this is something with broad learnability. In the process, it is interesting to see the manner in which the externality needs to make it into the payoffs and how that space needs to be structured.

Overall the paper is well written and has good fit at NeurIPS. Relative to prior work, it is able to show that as long as the upstream agent is using a suitable approach, any bandit learning algorithm will work. This is a much stronger guarantee for the downstream agent as they are less dependent on exactly what the upstream agent is doing.

**Weaknesses:**

The paper and situation is quite interesting, but is one specific (classic) example scenario so applicability is good because it is a classic example, but still limited. A natural followin question is how broad do we expect these situations to go - can we recover general externalities in multi agent situations?

**Questions:**

How broad do you expect the BELGIC routine to be applicable? In a general $M$ agent situation with less obvious structure on the externalities would you expect optimal transfers to still be learned efficiently?

**Limitations:**

Yes the authors have adequately addressed limitations of their work.

---

> ### Author Rebuttal · Authors · 2024-08-05
>
> We thank the reviewer for the detailed feedback. All the comments will be taken into account in the new version of the paper. You can find below our answers to each of your concerns.
>
> > It will be nice to see a discussion about possible notions of optimality of mechanisms that incentivize participation in this context.
>
> As explained in the general answer, with the current assumption on the upstream player, we cannot hope for a better bound than $T^{\frac{\kappa+1}{2}}$. Indeed, in the almost “ideal” scenario where the downstream player proposes a payment $\tau_{a}^{\star}+\varepsilon$ on the (socially) optimal arm $a$ at each round, within our assumptions, the upstream player might not pull $a$ a number of times $\frac{T^{\kappa}}{\varepsilon}$, while still having an upstream regret smaller than $T^{\kappa}$. This would then imply a total downstream regret of order $\frac{T^{\kappa}}{\varepsilon}+T\varepsilon$, which is minimized for $\varepsilon=T^{\frac{\kappa-1}{2}}$ with value $T^{\frac{\kappa+1}{2}}$. Note that this is also the reason for which we add an extra $T^{\frac{\kappa-1}{2}}$ term in the proposed payment in the final phase of the algorithm.
>
> Yet, it might be possible that stronger assumptions on the upstream player strategy (that remain realistic) could lead to better regret bounds in the end.
>
> > How broad do you expect the BELGIC routine to be applicable?
>
> We invite the reviewer to look at our general answer on the use of our work for more general settings. To summarize, if externalities go both ways (i.e. downstream action also influences upstream utility), the problem becomes much more complex and BELGIC shouldn’t learn the optimal equilibrium. This would then become a general repeated game, for which learning is hard and a long line of research already exists.
> Our goal in this work is instead to consider a simpler game structure, where learning is easier. Extending this kind of problem to more than two agents is an interesting question as explained in our general answer, and we believe that BELGIC can pave the way towards designing satisfying learning strategies in those games and understanding how they behave when increasing the number of involved agents.

---

### Official Review · Reviewer_rbRK · 2024-07-13

**Soundness:** 3
**Presentation:** 3
**Contribution:** 3
**Rating:** 6
**Confidence:** 4

**Summary:**

The paper explores the field of mitigating externalities in economic interactions by applying the Coase Theorem with hindsight rationality. The Coase Theorem, a fundamental concept in economics, suggests that in the presence of externalities, property rights, and bargaining strategies can be utilized to achieve an optimal outcome for all parties involved. However, traditional applications of the Coase Theorem assume that players have perfect knowledge of the game, which may not hold true in real-world scenarios.

In this paper, the authors introduce the concept of hindsight rationality, where players learn from past interactions to improve their decision-making processes. By incorporating hindsight rationality into the Coase Theorem framework, the paper aims to provide a mechanism through which players can adapt their strategies over time to maximize social welfare, even in the presence of uncertainty. The theoretical foundation of the paper is built upon a series of theorems supported by assumptions and proofs.

Overall, the paper offers a novel perspective on addressing externalities in economic settings by integrating hindsight rationality into the Coase Theorem framework. Through a rigorous theoretical analysis, ethical considerations, and a discussion of limitations, the paper contributes to the ongoing discourse on optimizing social welfare in the presence of externalities.

**Strengths:**

S1. One of the main strengths of the paper is its solid theoretical foundation. The authors create a clear framework based on well-known economic principles, like the Coase Theorem, and expand it by adding the idea of hindsight rationality. The thorough theoretical analysis, backed by detailed proofs and assumptions, strengthens the credibility and robustness of the proposed approach.

S2. By introducing the concept of hindsight rationality within the context of mitigating externalities, the paper offers a novel perspective on addressing economic inefficiencies.

**Weaknesses:**

W1. One significant weakness of the paper is the lack of empirical validation or experimental results to back up the proposed theoretical framework. Although the theoretical analysis is good, empirical evidence would improve the practical applicability and real-world relevance of the research findings.

W2. It would have been insightful to see a comprehensive discussion on the potential challenges and complexities associated with implementing the proposed approach in practical economic settings. Addressing implementation hurdles could provide valuable insights for policymakers and practitioners.

W3. The setting (extension of Coase theorem in a two-player bandit setting where the actions of one of the players affect the other player) considered in the paper is narrow and limited. The results are also exemplified for a simple setting of two firms, an upward and a downward. Do the observations apply to potentially broader settings?

W4. The mathematical notations provided in the paper sometimes become too difficult to follow and correlate with the text.

**Questions:**

Please clarify the points in the Weaknesses section.

**Limitations:**

Yes, the limitations of the proposed method are discussed in the paper.

---

> ### Author Rebuttal · Authors · 2024-08-05
>
> We thank the reviewer for the detailed feedback. All the comments will be taken into account in the new version of the paper. You can find below our answers to each of your concerns.
>
> > W1. One significant weakness of the paper is the lack of empirical validation or experimental results to back up the proposed theoretical framework.
>
> We will add in the revised version experiments on a toy model. Please see the attached pdf for the corresponding, unpolished, figure. In a few words, we consider a simple environment with two firms having continuous (quadratic) profit functions, similarly to Example 1. We discretize these functions so each level corresponds to an arm. We consider the two cases where (i) no property rights are defined and both firms playing UCB; (ii) property rights are defined, with the downstream firm applying BELGIC (and upstream plays UCB). In both cases, we plot the empirical frequency of the action chosen by the upstream firm.
> As expected by our theory, we observe in this example that the learning agents will quickly converge to the social optimum when using transfers and implementing the algorithm BELGIC (for the downstream player). On the other hand, there is a social inefficiency in the absence of transfers, as the upstream player ends up choosing his best action, regardless of the downstream utility.
>
> > W2. Addressing implementation hurdles could provide valuable insights for policymakers and practitioners.
>
> Addressing implementation of our approach on real world situations for policymakers and practitioners requires an extensive line of research that is generally run over multiple years in economics. We believe that studying these questions is out of our scope and left for future work. Before that, we also believe that building a more solid theoretical understanding of the problem is necessary, through extensions to more realistic cases (as in our answer below to W3).
>
> As an example, Abildtrup et al (2012) observed that the predictions of Coase theorem are not verified for an interaction between farmers and waterworks in Denmark. They suggested that the uncertainty on the reward functions might be a reason for that. We can see our work as an illustration of the cost of learning in the presence of uncertainty: this cost might be an obstacle for practical implementation.
>
>
> > W3. Do the observations apply to potentially broader settings?
>
> Please see our general answer on the use of our work for more general settings.
> To summarize, extending our work to broader settings is a very interesting direction for future work. We believe that our work can pave the way towards tackling more general settings.
>
> > W4. The mathematical notations provided in the paper sometimes become too difficult to follow and correlate with the text.
>
> We realize that we build on a quite heavy set of notations and have faced several times the challenge of writing something clear but still rigorous. The latter forced us to rely on several different mathematical notations but we can either introduce clearer environment definitions or a table of notations.
>
> If the reviewers agree, we would love to include a table of notations after the references to help the reader. We can also insist on the notations and their definitions each time we introduce a new one to ease the reading.
>
> --------
>
> We hope that we answered all your different concerns. If you still have any, we would be happy to answer any further questions.
>
> ---------
> **References**
>
> Abildtrup, J., Jensen, F., & Dubgaard, A. (2012). Does the Coase theorem hold in real markets? An application to the negotiations between waterworks and farmers in Denmark. Journal of environmental management, 93(1), 169-176.

---

> > ### Comment · Reviewer_rbRK · 2024-08-12
> >
> > Thank you for your responses. The authors address some of my concerns and I am happy to increase my score to 6.

---

### Official Review · Reviewer_aRFf · 2024-07-16

**Soundness:** 3
**Presentation:** 4
**Contribution:** 3
**Rating:** 7
**Confidence:** 2

**Summary:**

The paper presents a two-player sequential game with bandit feedback where one player's actions create externalities that affect the other player's outcomes. In this model, the utility of only one player (the downstream player) is impacted by the collective actions of both the upstream and downstream players. They consider two scenarios: one where players act independently to maximize individual utility, and another where players can interact via transfers.

In the first scenario (without transfers), the authors show that individual utility maximization can lead to suboptimal social welfare in some instances. In the second scenario, where the downstream player can choose a transfer, they demonstrate that social welfare can reach its optimum. They propose an algorithm for the downstream player that achieves this outcome, provided the upstream player follows any no-regret policy. This effectively presents an online version of the Coase theorem, establishing that even with imperfect information, the two-player sequential game converges to the global optimum outcome when transfers are allowed.

**Strengths:**

1. The algorithm does not assume a specific policy for the upstream player and achieves sublinear regret for any no-regret upstream policy (with regret scaling as $T^\kappa$).
2. Lemma 1 provides a clean problem formulation, showing that bounds on regret in individual utility (with transfers) lead to a bound on the social welfare regret.
3. For the specific instance where the upstream player follows the Upper Confidence Bound (UCB) policy (or any other algorithm with  $\sqrt{T}$ regret) the proposed algorithm achieves regret scaling as  $T^\frac{3}{4}$.

**Weaknesses:**

Please see the questions section.

**Questions:**

1. It is not clear whether the regret incurred by the downstream player is optimal. Even if the upstream player is assumed to follow UCB, it's not evident whether a regret scaling as $\sqrt{T}$ policy is achievable for the downstream player as well.
2. It is stated (on lines 111-112) that the assumption that the rewards are bounded is "without loss of generality" (WLOG). Can you please explain why? Would this work even in the case of other standard reward distributions in bandit literature, such as sub-Gaussian rewards?
3. In the current setup, it is assumed that transfers occur only on a single action, with the transfer function taking a value of 0 for all but one action. Is there a particular reason for assuming such a transfer function? Can we work with more general transfer functions where the downstream player can choose to make transfers on multiple actions? How would this affect the regret of the downstream player?

**Limitations:**

This paper does not have any negative social impact.

---

> ### Author Rebuttal · Authors · 2024-08-05
>
> Thank you for your insightful feedback, that will be taken into account in the new version of the paper. We answer below your different questions.
>
> > It is not clear whether the regret incurred by the downstream player is optimal.
>
> See our general answer on optimality of the regret bound for BELGIC for more details.
> To summarize, we believe that the $T^{\frac{1+\kappa}{2}}$ bound is optimal with our set of assumptions. Yet, it might be possible that stronger assumptions on the upstream player strategy (that remain realistic) could lead to better regret bounds in the end.
>
> > It is stated (on lines 111-112) that the assumption that the rewards are bounded is "without loss of generality"
>
> Note that our boundedness assumption is about the expected rewards $v^{\mathrm{up}}(a)$. On the other hand, we assume nothing about the random realization rewards, except 1-sub-Gaussianity in Corollary 1.
> H1 is only needed to know the range for which we should run the binary search (to estimate the optimal transfers $\pi^{\star}$). When the range is unknown, we could first do a range search procedure before the binary search, but we eluded this point for the sake of simplicity and presentation.
>
> > In the current setup, it is assumed that transfers occur only on a single action, with the transfer function taking a value of 0 for all but one action
>
> More generally, we could indeed define transfers as a vector of size $K$ depending on the choice of the upstream player. However such a possibility does not bring any improvement in the considered solution for the downstream player, as we cannot infer more information from the choice of action by the upstream player in that case.

---

> > ### Comment · Reviewer_aRFf · 2024-08-13
> >
> > Thank you for your response. I have no further questions.

---

### Author Rebuttal · Authors · 2024-08-05

We thank all the reviewers for their detailed and insightful feedback. All their comments will be taken into account in the revised version of our work. We answer individually to each reviewer's concern. Besides, a couple points were raised by multiple authors: we answer these points below.

## About the optimality of the regret bound for BELGIC

This is a very interesting question. With the current assumption on the upstream player, we cannot hope for a better bound than $T^{\frac{\kappa+1}{2}}$. Indeed, in the almost “ideal” scenario where the downstream player proposes a payment $\tau_{a}^{\star}+\varepsilon$ on the (socially) optimal arm $a$ at each round, within our assumptions, the upstream player might not pull $a$ a number of times $\frac{T^{\kappa}}{\varepsilon}$ times, while still having a regret smaller than $T^{\kappa}$. This would then imply a total regret of order $\frac{T^{\kappa}}{\varepsilon}+T\varepsilon$ to the downstream player, which is minimized for $\varepsilon=T^{\frac{\kappa-1}{2}}$ with value $T^{\frac{\kappa+1}{2}}$. Note that this is also the reason for which we add an extra $T^{\frac{\kappa-1}{2}}$ term in the proposed payment in the final phase of the algorithm.

Yet, better bounds might be reachable under stronger assumptions on the upstream regret (e.g., instance dependent bounds), which is open for future work.

## Can our algorithm be used for more general problems?

Naturally, if externalities go both ways (i.e., downstream action also influences upstream utility), the problem becomes much more complex and BELGIC shouldn’t learn the optimal equilibrium. This would then become a general repeated game, for which learning is hard and a long line of research already exists.

Our goal in this work is instead to consider a simpler game structure, where learning is easier. Extending our work to more than two agents is indeed a very interesting direction that we plan pursuing in future works. A potential extension of the setting could be for instance to consider a chain of agents, whose actions all influence the following agent in the chain.
A stronger set of assumptions would be needed in that case, as we would need to control the behaviors of all agents. We believe that our work can pave the way towards designing satisfying learning strategies in such games and it is a promising direction for future work.

---

### Decision · Program_Chairs · 2024-09-25

**Decision:**

Accept (spotlight)

**Comment:**

There was a consensus that this work tackles an interesting problem and provides significant results. All questions raised by the reviewers were addressed by the authors, and no further issues were raised during the discussion period. This is an easy accept (spotlight).